# Resveratrol modulates the inflammatory response via an estrogen receptor-signal integration network

Jerome C Nwachukwu[1], Sathish Srinivasan[1], Nelson E Bruno[1], Alexander A Parent[2], Travis S Hughes[3], Julie A Pollock[2], Olsi Gjyshi[1], Valerie Cavett[1], Jason Nowak[1], Ruben D Garcia-Ordonez[3], René Houtman[4], Patrick R Griffin[3], Douglas J Kojetin[3], John A Katzenellenbogen[2], Michael D Conkright[1], Kendall W Nettles[1]*

[1]Department of Cancer Biology, The Scripps Research Institute, Jupiter, United States; [2]Department of Chemistry, University of Illinois, Urbana, United States; [3]Department of Molecular Therapeutics, The Scripps Research Institute, Jupiter, United States; [4]Nuclear Receptor Group, PamGene International, Den Bosch, Netherlands

**Abstract** Resveratrol has beneficial effects on aging, inflammation and metabolism, which are thought to result from activation of the lysine deacetylase, sirtuin 1 (SIRT1), the cAMP pathway, or AMP-activated protein kinase. In this study, we report that resveratrol acts as a pathway-selective estrogen receptor-α (ERα) ligand to modulate the inflammatory response but not cell proliferation. A crystal structure of the ERα ligand-binding domain (LBD) as a complex with resveratrol revealed a unique perturbation of the coactivator-binding surface, consistent with an altered coregulator recruitment profile. Gene expression analyses revealed significant overlap of TNFα genes modulated by resveratrol and estradiol. Furthermore, the ability of resveratrol to suppress *interleukin-6* transcription was shown to require ERα and several ERα coregulators, suggesting that ERα functions as a primary conduit for resveratrol activity.

*For correspondence: knettles@scripps.edu

**Competing interests:** The authors declare that no competing interests exist.

**Reviewing editor**: Leemor Joshua-Tor, Cold Spring Harbor Laboratory, United States

## Introduction

Many beneficial effects on human health have been described for resveratrol ((E)-5-(*p*-hydroxystyryl) resorcinol), including prevention of skin and colorectal cancer, protection from metabolic and cardiovascular disease, neuroprotection, and general anti-inflammatory effects. Efficacy associated with resveratrol use has been attributed to activation of the lysine deacetylase, Sirtiun 1 (SIRT1) (*Baur and Sinclair, 2006*), the cAMP pathway, or the AMP-activated protein kinase (AMPK) (*Park et al., 2012*; *Price et al., 2012*; *Tennen et al., 2012*).

Resveratrol is also a phytoestrogen that modulates estrogen receptor (ER)-mediated transcription (*Gehm et al., 1997*; *Bowers et al., 2000*), though only a small percent of published papers consider ER as a potential mediator of the complex pharmacology of resveratrol. The estrogenic role of resveratrol is important because a variety of resveratrol-sensitive tissues are ER-positive, and the two ER subtypes in mammals, ERα and ERβ, exhibit different tissue-specific expression profiles (*Bookout et al., 2006*). Specifically, effects of resveratrol on ER include anti-inflammatory effects such as protection from trauma-hemorrhage-induced injury and suppression of *Interleukin-6* (*IL-6*) expression in the liver, intestine, and cardiovascular system (*Yu et al., 2008*, *2010*, *2011b*). However, in contrast to other ERα agonists, resveratrol does not induce proliferation of mammary or uterine tissues (*Turner et al., 1999*), allowing it to be taken as a dietary supplement. The structural and molecular mechanisms for this pathway-selective signaling are not known.

**eLife digest** Resveratrol is a compound found in significant quantities in red wine, grapes, and peanuts. Many health benefits have been linked to it, including protecting against certain types of cancer and reducing the risk of cardiovascular disease. How resveratrol could produce these very different effects is unknown, but evidence is emerging that it is involved in a wide range of biological processes.

However, the ability of resveratrol to bind with, and activate, proteins called estrogen receptors has largely been overlooked. These receptors have a range of roles. For example, estrogen receptors fight against inflammation by preventing the transcription of the gene that encodes a signaling protein called interleukin-6. However, estrogen receptors do not work alone: other molecules called coregulators interact with them and alter how effectively they can prevent gene expression.

Resveratrol has also been associated with anti-inflammatory effects, particularly in tissues that contain large numbers of an estrogen receptor called ERα, though this connection has been little studied. Nwachukwu et al. now reveal that the two are linked—the anti-inflammatory response of resveratrol relies on it being bound to ERα. This binding changes the shape of the receptor in a way that controls which coregulator molecules help it to regulate transcription. Additionally, this binding complex does not produce the cancer-causing side effects often associated with activated ERα. Nwachukwu et al. also found that the effect of resveratrol on the inflammatory response depends on other specific coregulators being present.

The role of ERα in enhancing and activating resveratrol's effects is important because resveratrol has poor bioavailability in humans, and so it is not easily absorbed into the bloodstream. This makes it difficult for someone to get a dose high enough to produce beneficial effects. Further research targeting ERα may produce similar beneficial compounds to resveratrol, but with improved bioavailability.

The roles of resveratrol as a stimulant of SIRT1 and ER signaling have been presented as distinct mechanisms. However, dissection of these mechanisms of action is complicated by physical and functional interactions between ERα and SIRT1, where: (i) ERα is a SIRT1 substrate (*Kim et al., 2006*; *Ji Yu et al., 2011*), and (ii) SIRT1 functions as an ER coregulator required for the oncogenic effects of estrogens in breast cancer (*Elangovan et al., 2011*). Further, SIRT1 also deacetylates NF-κB subunits to inhibit expression of inflammatory genes (*Rothgiesser et al., 2010*), and ERα also inhibits NF-κB signaling (*Cvoro et al., 2006*; *Nettles et al., 2008a, 2008b*; *Saijo et al., 2011*; *Srinivasan et al., 2013*). Thus, understanding the anti-inflammatory actions of resveratrol requires careful dissection of its ER-mediated vs non ER-mediated effects, and the role of SIRT1.

ER activates transcription in response to estradiol (E2), and a wide cast of other estrogenic compounds, including steroids, phytoestrogens, and environmental estrogens, by either direct binding to DNA, or tethering to DNA-bound transcription factors (*Cicatiello et al., 2004*; *DeNardo et al., 2007*). Transactivation via direct binding of ER to estrogen response elements (EREs) has been well studied, and it involves ER-mediated recruitment of transcriptional coregulators, including coactivators and corepressors (*Shang et al., 2000*; *Metivier et al., 2003*). These coregulators remodel chromatin, regulate post-translational modification (PTM) of histones and non-histone substrates, and control assembly of transcription-initiation and transcription-elongation complexes at target gene promoters (*Bulynko and O'Malley, 2011*; *Perissi et al., 2010*). Coregulator function in ER-mediated transcription is consistent with a *hit-and-run* model where one coregulator complex lays down PTMs and changes the chromatin and coregulator environment so as to increase affinity for the next coregulator complex (*Shang et al., 2000*; *Fletcher et al., 2002*; *Metivier et al., 2003*).

In contrast, ER-mediated repression of inflammatory genes has been less extensively studied. ER represses transcription through a tethering mechanism called transrepression, via interaction with NF-κB and activator protein-1 complexes. Only a few key coregulators involved in this process have been identified (*Cvoro et al., 2006*; *Yu et al., 2007*; *Nettles et al., 2008b*; *Saijo et al., 2011*). Moreover, the mechanism through which resveratrol modulates the inflammatory response is poorly understood. In a screen for ERα ligands that inhibit IL-6 production, we found that resveratrol was among the most

efficacious (*Srinivasan et al., 2013*), prompting us to explore this mechanism. To address the question of how resveratrol regulates *IL-6* without stimulating proliferation, we examined the roles of ERα, SIRT1, and a cast of coregulators. Resveratrol inhibited *IL-6* expression via ERα, which was recruited to the *IL-6* promoter where it altered the recruitment profile of coregulators, including SIRT1, and reduced acetylation of p65 NF-κB, which is required for transcriptional activation. Unexpectedly, there was a marked diversity of coregulators required for signal integration, where many display distinct roles in TNFα versus ERα signaling.

## Results

### Resveratrol is a pathway-selective ERα ligand

Resveratrol, which has a non-steroidal chemical structure (*Figure 1A*), profiled as a partial agonist in ER-positive MCF-7 breast cancer cells, stimulating 3xERE-luciferase reporter activity with about 30% efficacy relative to E2 (*Figure 1B*). To assess the effect of resveratrol on MCF-7 cell proliferation, cells in steroid-depleted media were treated for 7 days with several ER ligands including resveratrol. Unlike E2, resveratrol did not stimulate cell proliferation (*Figure 1C*).

Steroid receptor coactivators, SRC1, SRC2, and SRC3, are primary mediators of ERα activity, and they provide a scaffold for recruitment of other coregulators such as p300 and CBP (*Chen et al., 2000*; *Wong et al., 2001*; *Huang and Cheng, 2004*). Despite their overlapping functions, SRCs play disparate roles in normal mammary gland development, with SRC3, and to some extent SRC1, contributing to growth (*Xu and Li, 2003*). In MCF-7 cells, SRC3 is selectively required for E2-induced proliferation (*Karmakar et al., 2009*). When compared to E2, resveratrol induced full association of ERα with SRC2, but reduced interaction with SRC1 or SRC3 in a mammalian two-hybrid assay (*Figure 1D*), which we propose, is not a sufficient interaction to support the proliferative response. This idea is further supported by chromatin immunoprecipitation (ChIP) assays examining recruitment of these factors to a canonical ERα binding site in the *GREB1* gene, a gene required for estrogen-induced cell proliferation (*Rae et al., 2005*; *Sun et al., 2007*). We found that resveratrol induced less SRC3 recruitment than was observed upon E2 treatment, but induced comparable SRC2 and ERα recruitment (*Figure 1E,F*). Thus, the lack of proliferative signal is consistent with ligand-selective coregulator recruitment by resveratrol-bound ERα and the disparate roles of the SRCs in the proliferative response. Together with anti-inflammatory effects described below, these results indicate that resveratrol acts as a pathway-selective ERα agonist.

### Resveratrol modulates the inflammatory response through ERα

ERα coordinates a wide range of physiologic events outside of reproductive tissues, including modulation of brain function, cardiovascular and bone health, metabolic functions in the liver and muscle, remodeling of the immune system, and coordination of the inflammatory response in ERα-target tissues (*Nilsson et al., 2011*). TNFα or Toll-like receptor agonists such as lipopolysaccharide (LPS) trigger rapid translocation of NF-κB transcription factors from the cytoplasm into the nucleus, causing activation of inflammatory genes such as *IL-6* via direct binding of NF-κB to κB response elements, recruitment of transcriptional coactivators, and assembly of transcription-initiation and transcription-elongation complexes at target gene promoters (*Ben-Neriah and Karin, 2011*). ER-mediated suppression of inflammatory genes can occur by inhibition of NF-κB translocation or DNA binding, or through a transrepression mechanism involving recruitment of ERα to the cytokine promoters via protein–protein interactions (*Ghisletti et al., 2005*; *Cvoro et al., 2006*; *Nettles et al., 2008b*; *Saijo et al., 2011*), a mechanism that is also evident with anti-inflammatory effects of the glucocorticoid receptor (*Uhlenhaut et al., 2013*).

For detailed mechanistic studies, we decided to focus on *IL-6*, whose suppression by ERα ligands in MCF-7 cells has remained robust and consistent over time (*Srinivasan et al., 2013*), unlike others genes such as *monocyte chemoattractant protein-1 (MCP-1)*, whose inhibition has been variable (not shown). Treatment of MCF-7 cells with TNFα increased secretion of IL-6 protein, and E2 or resveratrol inhibited this response (*Figure 2A*). The full ERα antagonist, faslodex/fulvestrant/ICI 182,780 (ICI) reverses resveratrol-dependent inhibition of IL-6 production by these cells (*Srinivasan et al., 2013*); thus ERα mediates resveratrol-directed inhibition. Similar ERα-mediated effects were observed in mouse RAW2645.7 macrophages stimulated with LPS (*Figure 2B*), which again were reversed by ICI. To fully characterize the role of resveratrol and ER in coordinating the inflammatory response, MCF-7

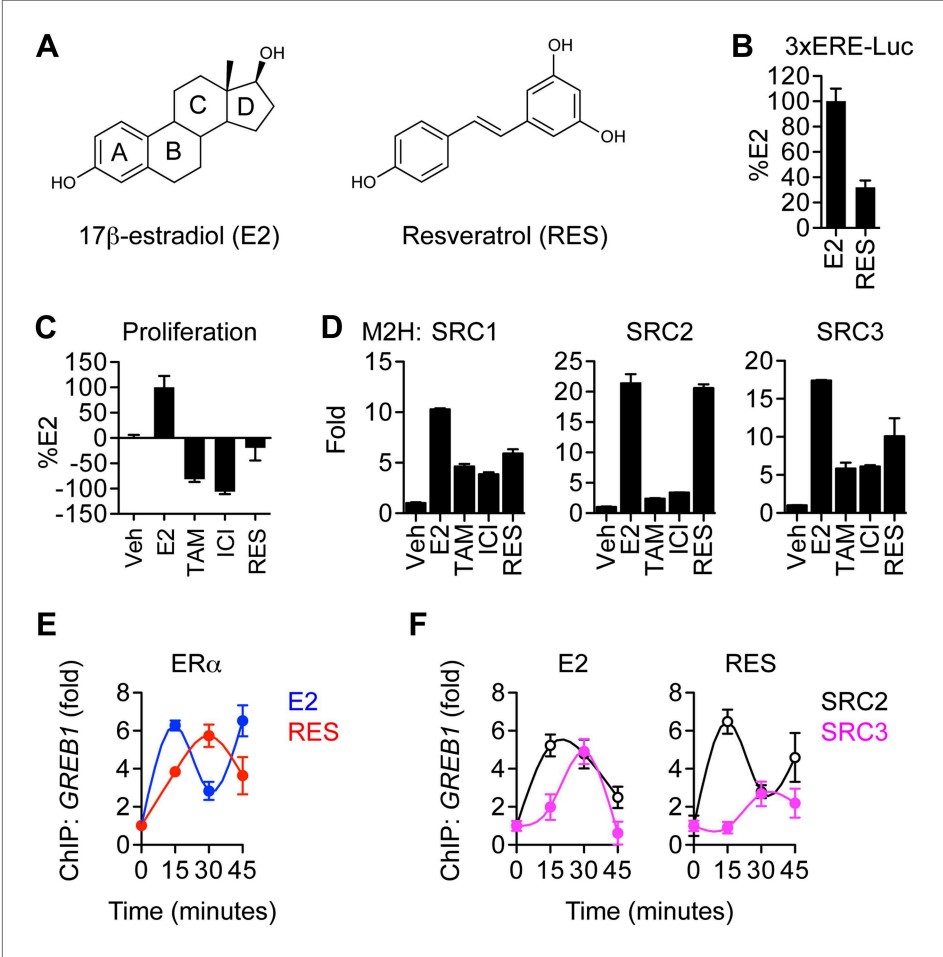

**Figure 1**. Effects of resveratrol on the canonical ERα proliferative pathway. (**A**) Chemical structures of E2 and resveratrol. (**B**) Luciferase assay of MCF-7 cells transfected with 3xERE-luciferase reporter and stimulated with 10 nM E2 or 10 µM resveratrol. (**C**) Steroid-deprived MCF-7 cells were treated with 10 nM E2, 10 µM 4-hydroxytamoxifen (TAM), 10 µM ICI182, 780, or 10 µM resveratrol. After 7 days, cell number was determined with a standard curve. (**D**) Mammalian two-hybrid assays with ERα and the coactivators SRC1-3. HEK293-T cells were transfected with Gal4 SRC1-3 fusions, ERα-VP-16, and the 5xUAS-luciferase reporter for 24 hr. Cells were treated with 10 nM E2, 10 µM TAM, 10 µM ICI, or 10 µM resveratrol for 24 hr and processed for luciferase activity. Data are presented in panels **B**–**D** as mean ± SEM. (**E**) Resveratrol-induced recruitment of ERα to the *GREB1* promoter. Occupancy of *GREB1* by ERα was compared by ChIP assay in MCF-7 cells that were steroid deprived for 3 days, treated with 10 nM E2 or 10 µM resveratrol, and fixed after 0, 15, 30, or 45 min (mean ± SEM *n* = 2). (**F**) Resveratrol reduced SRC3 but not SRC2 recruitment at the *GREB1* promoter. Occupancy of *GREB1* by SRC2 and SRC3 were examined by ChIP assay in MCF-7 cells treated as described in panel **A**. Average promoter occupancies are shown as fold changes (mean ± SEM *n* = 2).

cells were treated with TNFα and either E2 or resveratrol, and gene expression was analyzed using Affymetrix cDNA microarrays. Notably, almost all of the resveratrol-modulated genes were also E2 regulated (***Figure 2C,D***), supporting an ER-mediated mechanism of action. Interestingly, genes that were modulated by ERα ligand in the same direction as TNFα were more sensitive to E2 than resveratrol (***Figure 2C***). In contrast, resveratrol had a greater impact in opposing TNFα activity (***Figure 2D***). The set of genes that were regulated in opposite directions at least twofold by E2 vs resveratrol was less than 0.5% of total; thus nearly all of the effects of resveratrol were ERα-mediated in this context.

Confirmation of the ligand-modulated, TNFα-induced gene expression profile with qPCR showed that *IL-6*, *prostaglandin E receptor 4* (*PTGER4*), and *TNF receptor superfamily member 11b* (*TNFRSF11B*) were TNFα-induced, and equally suppressed by E2 or resveratrol (***Figure 2E***). Importantly, the effects of resveratrol on expression of these inflammatory genes were fully reversed by ICI in both MCF-7

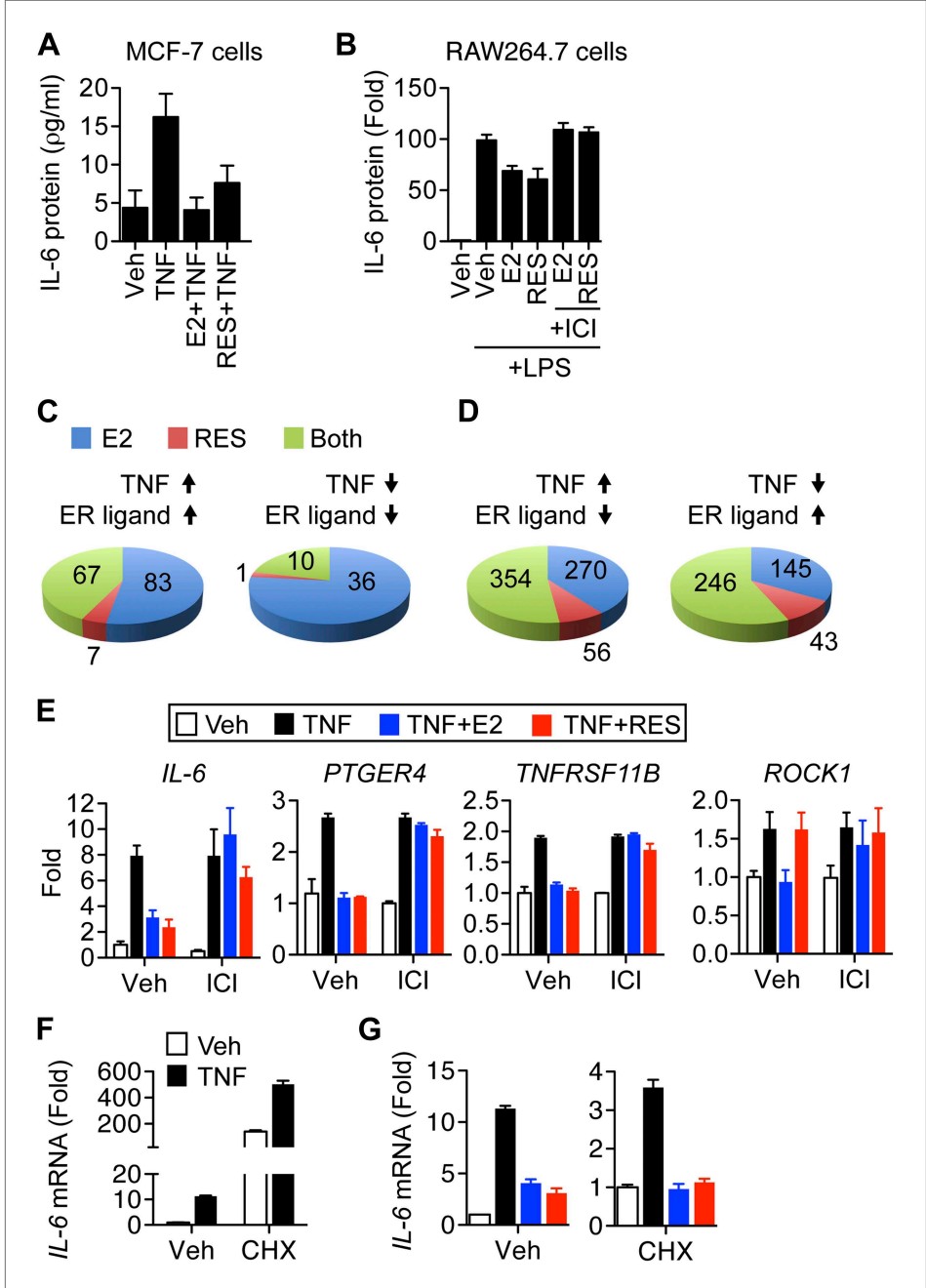

**Figure 2**. Resveratrol represses inflammatory genes through ER. (**A**) MCF-7 cells were plated into charcoal-stripped phenol red free media and treated for 24 hr with 1 ng/ml TNFα ±10 µM E2, or 10 µM resveratrol. Secreted IL-6 protein was measured from the media using AlphaLISA. Mean ± SEM of biological triplicates are shown. (**B**) RAW264.7 macrophages were treated as in panel **A**, and stimulated with LPS as indicated. Mean ± SEM from biological triplicates are shown. (**C** and **D**) Steroid-deprived MCF-7 cells were treated for 4 hr with 10 ng/ml TNFα alone or in combination with 10 nM E2 or 10 µM resveratrol. Total RNA was reverse transcribed and analyzed using Affymetrix Genechip microarrays. Transcripts showing ≥twofold changes in expression upon TNFα stimulation were classified as indicated. Summary of genes regulated (**C**) in the same direction or (**D**) in opposite directions by TNFα and ER ligands are shown. (**E**) Steroid-deprived MCF-7 were pre-treated for 1 hr with ethanol vehicle or 1 µM ICI, and then treated as indicated with 10 ng/ml TNFα, 10 nM E2, and 10 µM resveratrol for 2 hr. Total RNA reverse-transcribed and analyzed by qPCR for the indicated mRNAs. Mean ± SEM of a representative experiment of biological duplicates are shown. (**F** and **G**) IL-6 mRNA levels in steroid-deprived MCF-7 cells pre-treated with vehicle or 10 µg/ml CHX for 1 hr and stimulated TNFα, E2, and resveratrol as in panel **D** for 3 hr were analyzed by

*Figure 2. Continued on next page*

*Figure 2. Continued*

qPCR. Levels in the control samples (first bar) of each graph were arbitrarily set to 1. Mean ± SEM of a representative experiment are shown.

The following figure supplements are available for figure 2:

**Figure supplement 1**. Resveratrol represses *IL-6* in a dose-dependent manner.

**Figure supplement 2**. Resveratrol represses inflammatory genes through ER.

**Figure supplement 3**. Resveratrol represses *IL-6* in cycloheximide-treated cells.

(*Figure 2E*, *Figure 2—figure supplement 1*), and T47D breast cancer cells (*Figure 2—figure supplement 2*), demonstrating that ERα mediates resveratrol-dependent repression of these genes. Other genes such as *Rho-associated, coiled-coil containing protein kinase 1* (*ROCK1*) exhibited E2-selective repression (*Figure 2E*), consistent with the array data showing some E2-selective genes.

To determine if resveratrol and E2 repress *IL-6* indirectly, via transcriptional regulation of another protein that regulates NF-κB activity (*Auphan et al., 1995*; *Scheinman et al., 1995*; *King et al., 2013*), cells were pre-treated with vehicle or the protein-synthesis inhibitor, cycloheximide (CHX). In both MCF-7 and T47D cells, CHX led to super-induction of *IL-6* mRNA (*Figure 2F*, *Figure 2—figure supplement 3*), which is a hallmark of CHX response (*Faggioli et al., 1997*; *Hershko et al., 2004*). However, CHX did not affect repression of TNFα-induced *IL-6* expression by E2- or resveratrol (*Figure 2G*, *Figure 2—figure supplement 3*). Thus, resveratrol and E2 do not require de novo protein synthesis for this repression. Collectively, these results suggest that resveratrol modulates the inflammatory response through a direct, ERα-mediated transrepression mechanism, which we further verify with ChIP assays, below.

## Resveratrol alters the AF2 surface of ERα

Upon agonist binding, the ERα LBD undergoes a conformational change that allows helix 12 to dock across helix 11 and helix 3 (*Figure 3—figure supplement 1*), thereby forming a coactivator-binding surface called activation function 2 (AF2) (*Brzozowski et al., 1997*; *Shiau et al., 1998*; *Warnmark et al., 2002*). Importantly, removal of helix 12 from this position reveals a longer groove that binds an extended peptide motif found in transcriptional corepressors, such as NCoR and SMRT (*Heldring et al., 2007*). Further, antagonists can reposition helix 12 out of the active conformation, and stimulate recruitment of corepressors to this extended groove, or position helix 12 to block both coactivators and corepressors to the AF2 surface (*Shiau et al., 1998*; *Figure 3—figure supplement 1*).

By binding to the LBD, ER ligands may also facilitate recruitment of coactivators to another major coregulator-binding site in the unstructured amino-terminal domain of ERα, called AF1 (*Webb et al., 1998*; *Nettles and Greene, 2005*). In fact, the agonist activity of tamoxifen is mediated by AF1 in tissues with higher expression of coactivators that bind preferentially to that region (*McInerney and Katzenellenbogen, 1996*; *Shang and Brown, 2002*). These different potential signaling mechanisms were reviewed in *Nettles and Greene (2005)*.

The DNA-binding domain also contributes to AF2-mediated receptor activity through unknown mechanisms, further complicating matters (*Meijsing et al., 2009*; *Srinivasan et al., 2013*). In addition, coactivator recruitment to AF2 is also affected by partial agonists, which subtly reposition helix 11 to disrupt proper docking of helix 12 in its active position (*Nettles et al., 2008a*). Thus the AF2 surface represents a nexus for ligand-mediated control of both recruitment of coregulators to the LBD and allosteric signaling to other domains.

To define the structural basis for the selective anti-inflammatory property of resveratrol, the ERα LBD was crystallized in complex with resveratrol and the SRC2 nuclear receptor-interacting domain peptide containing an LxxLL motif (*Figure 3A*, *Table 1*). Unlike E2, which binds in a single orientation (*Brzozowski et al., 1997*; *Warnmark et al., 2002*; *Figure 3B*), resveratrol binds to ERα in two different orientations in one subunit of the dimer, shown as conformers #1 and #2 (*Figure 3C*). Conformer #1 shows the canonical *para* phenol of resveratrol mimicking the A-ring of E2, whereas in conformer #2, this is flipped. Also unexpectedly, in the other subunit of the dimer, resveratrol bound predominantly with the resorcinol group mimicking the A-ring of E2, in conformer #2. To our

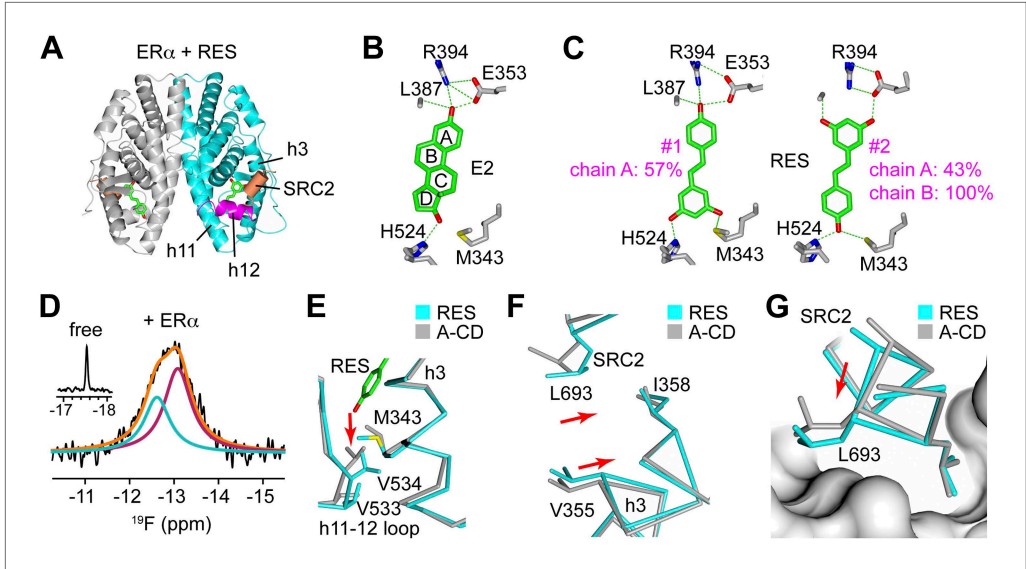

**Figure 3**. ERα adopts a resveratrol-specific conformation. (**A**) Crystal structure of ERα LBD in complex with resveratrol. The LBD is shown as a ribbon diagram with one monomer colored gray and the other cyan, except for helix 12 (h12), colored magenta. The receptor-interacting peptide of SRC2 (coral tube) docks at the AF2 surface. (**B**) Structure of E2-bound ERα shows that the A-ring forms a hydrogen-bonding network that is conserved among steroid receptors. PDB ID: 1ERE. (**C**) Binding orientations of resveratrol. Compared to E2, resveratrol binds in two distinct orientations. Conformer #1 shows the expected binding orientation, with the phenol mimicking the A-ring of E2. In contrast, the 'flipped' conformer #2 with the resorcinol mimicking the A-ring of E2 was unexpected and predominant. Hydrogen bonds (dashes) and residues that contact the resveratrol molecule are shown. (**D**) $^{19}$F-NMR of F-resveratrol. The inset shows a narrow peak in the spectrum of F-resveratrol in buffer (half-height line width = 27 Hz), while the broad peak for F-resveratrol bound to ERα LBD (modeled in orange) fits best to two NMR resonances (colored red and blue), consistent with two distinct binding modes. (**E–G**) Crystal structure of the ERα LBD in complex with the control compound i.e., an A-CD ring estrogen (gray), was superposed on the resveratrol-bound structure (cyan). In panel **E**, resveratrol (green) shifts h3 Met343 to disrupt the normal packing of the h11–h12 loop, shifting the position of V534 by 2.5 Å. In panel **F**, resveratrol-induced shift in h3 is transmitted allosterically via ERαV355 and ERα I358 to SRC2 L693 within its $^{690}$LxxLL$^{694}$ motif. Panel **G** shows the resveratrol-induced rotation of the SRC2 peptide.

The following figure supplements are available for figure 3:

**Figure supplement 1**. Crystal structures of the ERα LBD in complex with E2 and 4-hydroxytamoxifen (TAM).

**Figure supplement 2**. Chemical structures of F-resveratrol and the A-CD ring estrogen used as a structural control.

**Figure supplement 3**. Deconvolution of NMR signal from F-resveratrol bound to ERα.

**Figure supplement 4**. F-RES also binds ERα in two orientations.

**Figure supplement 5**. Electron density maps of resveratrol and F-resveratrol within the ERα ligand-binding pocket.

**Figure supplement 6**. Electron density maps of SRC2 peptides docked at the AF2 surface.

knowledge, this is the first example of a ligand-bound ER structure that does not have a *para* phenol moiety in that position.

We previously described the binding of ligand in multiple poses as 'dynamic ligand binding', as it was associated with a ligand's ability to stabilize different conformations of ERα. In this model, a dynamically binding ligand perturbs the conformational ensemble such that there are discrete populations of stable conformers, each associated with a specific binding pose, where each receptor can undergo a conformational change as it re-binds the ligand. Further, we showed that this phenomenon

**Table 1.** Data collection and refinement statistics for new ERα structures

| Ligand | Resveratrol | F-resveratrol | A-CD ring estrogen |
|---|---|---|---|
| PDB ID | 4PP6 | 4PPP | 4PPS |
| Data collection | | | |
| Space group | P 1 21 1 | P 1 21 1 | P 1 21 1 |
| a, b , c (Å) | 56.04, 84.67, 58.42 | 54.19, 81.93, 58.47 | 56.11, 84.19, 58.48 |
| α, β , γ (°) | 90.0, 108.32, 90.0 | 90.0, 110.86, 90.0 | 90.0, 108.35, 90.0 |
| Resolution (Å) | 33.7–2.2 (2.28–2.20) | 46.3–2.7 (2.78–2.69) | 33.5–1.9 (2.00–1.93) |
| Number of reflections | 22,678 (944) | 11,884 (481) | 38,369 (3443) |
| I/σ | 12.6 (2.9)* | 22.5 (1.7) | 27.7 (2.1) |
| $R_{merge}$ | 0.07 (0.21) | 0.09 (0.45) | 0.05 (0.45) |
| Completeness (%) | 86.14 (36.35) | 88.46 (35.98) | 98.68 (89.10) |
| Multiplicity | 2.5 (1.5) | 6.3 (5.8) | 3.5 (2.0) |
| Refinement | | | |
| Number of non-H atoms | | | |
| Protein | 3840 | 3710 | 4014 |
| Ligands | 51 | 54 | 36 |
| Water | 307 | 36 | 323 |
| $R_{work}/R_{free}$ | 16.79/22.22 | 18.38/23.90 | 17.38/20.15 |
| Ramachandran favored (%) | 99 | 95 | 98 |
| Ramachandran outliers (%) | 0.21 | 1.1 | 0 |
| Wilson B-factor | 17.31 | 44.12 | 27.03 |
| Average B-factor | | | |
| All atoms | 26.7 | 66.1 | 36.1 |
| Protein | 26.4 | 66.4 | 36.1 |
| Water | 29.7 | 42.9 | 40.4 |
| RMS deviations | | | |
| Bond lengths (Å) | 0.008 | 0.011 | 0.002 |
| Bond angles (°) | 1.03 | 1.26 | 0.61 |

*(Highest-resolution shell).

is a structural mechanism for partial agonist activity (*Nettles et al., 2008a*; *Bruning et al., 2010*; *Srinivasan et al., 2013*), which has now also been shown with a G protein-coupled receptor (*Bock et al., 2014*). Lastly, ERα ligands that exhibit this so called dynamic binding profile showed greater anti-inflammatory activity than matched controls that bound in a single pose (*Srinivasan et al., 2013*).

To assess whether this dynamic ligand binding occurs in solution, we used F19 NMR, which established dynamic ligand binding to PPARγ (*Hughes et al., 2012*) and ERα (*Srinivasan et al., 2013*). Our previous work established that fluorinated ligands display the expected line broadening in F19 NMR signal upon binding to proteins. However, there were characteristic differences in matched isomeric ligands that bound in either a single orientation, or multiple orientations to their respective proteins. The ligands that bound in a single orientation displayed a single broadened peak, while the ligands that bound in more than one orientation displayed either multiple broadened peaks, or a single, asymmetrically shaped peak that was best modeled as two overlapping peaks (*Hughes et al., 2012*; *Srinivasan et al., 2013*). Here, we synthesized resveratrol with a fluorine substitution at the *meta* position on the phenol to generate F-resveratrol (*Figure 3—figure supplement 2*), and examined binding of F-resveratrol to ERα. F-resveratrol alone showed a single sharp peak; however when bound to ERα, it displayed a very broad peak that fit best to two peaks (*Figure 3D*, *Figure 3—figure supplement 3*), indicating multiple binding modes. This dynamic binding was corroborated by the crystal structure of an F-resveratrol-bound ERα complex (*Table 1*)

that was best fit with two ligand-binding orientations similar to those displayed by resveratrol (*Figure 3—figure supplements 4, 5*).

The crystal structure of the ERα LBD in complex with an A-CD ring estrogen (*Figure 3—figure supplement 2*), which has the typical phenolic A-ring but like resveratrol, does not have an adjacent B-ring, showed the same space group and crystal packing as the resveratrol-bound ERα structure (*Table 1*). This was therefore used as a control agonist structure. Compared to the typical phenolic A-ring of the A-CD ring estrogen, the resorcinol group of resveratrol induced a shift in helix 3 via a hydrogen bond with the backbone of Leu387 (*Figure 3C*). In turn, the shift in helix 3 disrupts the loop that connects helix 11 to helix 12, which in solution should destabilize helix 12 in the agonist conformation. This impact on helix 12 is visualized by the 2.5 Å shift in the positioning of the γ-carbons of Val534 (*Figure 3E*). Helix 12 does not participate in crystal packing, so this change is ligand driven.

Notably, the shift in helix 3 also alters binding of the SRC2 peptide at the AF2 surface. Leu693 of the SRC2 peptide binds helix 3 between Val355 and Ile358, and is shifted by 1.6 Å in the resveratrol-bound structure (*Figure 3F*), inducing an overall rotation of the peptide (*Figure 3G*). The electron density for the peptides allowed clear visualization of this rotation (*Figure 3—figure supplement 6*). Here, the coactivator peptide participates in crystal packing, but the crystals are in the same space group and show the same crystal packing interactions. Thus the rotation of the peptide occurs despite being held in place by an adjacent molecule. In summary, resveratrol induced several unique structural perturbations in ERα, including shifts in the helix 11–12 loop, which should modulate helix 12 dynamics, and direct remodeling of the coactivator-binding surface, which could contribute to an altered receptor–coactivator interaction profile and the lack of a proliferative signal.

## Resveratrol alters coregulator peptide-binding at the AF2 surface of ERα

To determine if the resveratrol-induced structural changes at the AF2 surface directly affect binding of SRC peptides to the ERα LBD in vitro, a non-competitive FRET assay was performed using a fixed amount of GST-ERα LBD and ligands, and increasing doses of fluorescein-tagged SRC peptides. For each SRC family protein, the highest affinity LxxLL motif peptide from SRC1, SRC2, or SRC3, for the E2-bound ERα LBD complex, was selected for further comparisons (*Figure 4—figure supplement 1*). Although the $EC_{50}$ of the SRC2 peptide could not be determined accurately due to a lack of plateau on the curve, the $EC_{50}$s of all three SRC peptides were comparable for the resveratrol-bound ERα LBD (*Figure 4A*). This suggests that the coactivator-selectivity profile of resveratrol-bound ERα requires important regions outside the ERα LBD and SRC peptides.

To test if the altered AF2 surface was also apparent in solution, we analyzed ligand-induced binding of over 150 distinct, nuclear receptor-interacting, coregulator peptide motifs to the ERα LBD, including those derived from both coactivators and corepressors, using the microarray assay for real-time coregulator-nuclear receptor interaction (MARCoNI) (*Aarts et al., 2013*). Hierarchical clustering of the peptide-binding results showed that compared to E2, resveratrol showed similar patterns of recruitment, but with reduced binding of most coactivator peptides to the ERα LBD (*Figure 4B*, *Figure 4—figure supplement 2*). However, there was a subset of peptides that were dismissed by E2, including several from the NCoR corepressor, which resveratrol failed to dismiss. In contrast, the A-CD ring estrogen and E2 had similar effects on coactivator peptide recruitment to, or dismissal from the ERα LBD (*Figure 4C*, *Figure 4—figure supplement 2*), consistent with a fully functional AF2 surface. Taken together, these results suggest that resveratrol binds the ERα LBD, and induces an altered AF2 surface, which reduces affinity for most peptides, but enables selectivity in the context of full-length receptor and coregulators, as shown by our mammalian two hybrid and ChIP data.

## Multifactorial control of ERα and TNFα signaling to the *IL-6* gene

ERα uses a large array of coregulators to activate transcription (*Bulynko and O'Malley, 2011*; *Lupien et al., 2009*; *Metivier et al., 2003*), but much less is known about the requirements for ERα-mediated transrepression. To identify factors required for E2 and resveratrol-dependent repression of *IL-6*, we undertook a small-scale siRNA screen, targeting over 25 factors including estrogen receptors and known ERα-interacting coregulators. ERα knockdown blocked inhibition of *IL-6* expression by both E2 and resveratrol, unlike siRNA against ERβ or the estrogen-binding G protein-coupled receptor, GPR30 (*Figure 5A*, *Figure 5—figure supplements 1, 2*), confirming that ERα mediates both E2- and resveratrol-dependent repression of *IL-6*.

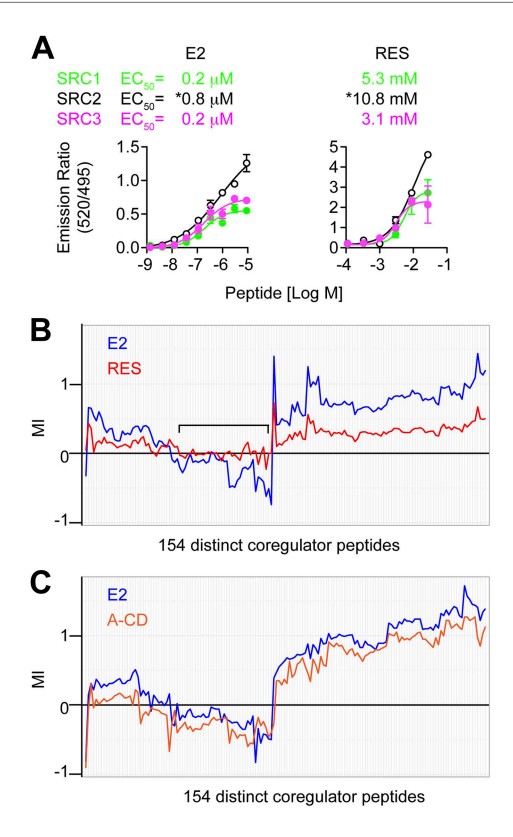

**Figure 4**. Resveratrol alters the binding of coregulator peptides to the ERα LBD. (**A**) E2- and resveratrol-induced binding of SRC1, SRC2, and SRC3 peptides to the ERα LBD were compared using LanthaScreen assay performed at fixed ligand concentrations, with increasing doses of SRC peptides. Mean ± SEM ($n$ = 3) are shown. *$EC_{50}$ could not be determined accurately since the saturating SRC2 peptide dose is unclear. (**B** and **C**) Hierarchical clustering of coregulator peptide-binding at the AF2 surface induced by 1 µM E2, (**B**) 100 µM resveratrol or (**C**) 1 µM A-CD ring estrogen, was performed using the quantitative in vitro assay, MARCoNI. MI >0 suggests ligand-induced recruitment, while MI <0 suggests ligand-dependent dismissal of a peptide compared to DMSO vehicle. The black bracket shows a cluster of E2 dismissed peptides that are not dismissed by resveratrol. See *Figure 4—figure supplement 2* for more details.

The following figure supplements are available for figure 4:

**Figure supplement 1**. SRC peptides.

**Figure supplement 2**. Details of proteomic comparison of ligand-induced binding of coregulator peptides using MARCoNI.

Knockdown of SRC1, SRC2, and SRC3 by RNA-interference revealed that these coregulators play distinct but overlapping roles in controlling *IL-6* expression. SRC1 and SRC3 knockdown led to an increase in *IL-6* mRNA in cells treated with either vehicle or TNFα, indicating a general role in repressing *IL-6* transcription (*Figure 5B*). SRC3 knockdown also blocked E2- and resveratrol-mediated suppression, suggesting an additional role for SRC3 in integrating TNFα and ERα signaling. By contrast, SRC2 knockdown markedly reduced TNFα-directed induction of *IL-6* transcripts, demonstrating that it is required for coactivation of TNFα induction of this gene. However, SRC2 knockdown also demonstrated that SRC2 is required for repression of *IL-6* by E2 or resveratrol (*Figure 5B*), suggesting that these ER ligands switched SRC2 function from that of a coactivator to a corepressor. This is similar to the context-dependent role of SRC2/GRIP1 in glucocorticoid action (*Rogatsky et al., 2002*), and the gene-specific role of silencing mediator for retinoid and thyroid hormone receptors (SMRT/NCOR2), which corepresses some ERα-target genes, while being required for activation of others (*Peterson et al., 2007*).

Knockdown studies also established that the acetyltransferase, CBP, was also required for suppression of *IL-6* induction by E2 and resveratrol, whereas p300 and pCAF were rather required for TNFα-induced expression of *IL-6* (*Figure 5C*), suggesting that CBP and p300 play opposing roles in repression vs activation of the same gene. Finally, knockdown of macro domain protein 1 (LRP16), a coactivator for both ERα and NF-κB (*Han et al., 2007*; *Wu et al., 2011*), also dampened the TNFα response, but did not affect suppression of *IL-6* by either E2 or resveratrol (*Figure 5—figure supplement 1*).

We also tested the roles of various components of complexes that harbor dedicated corepressors, including nuclear receptor corepressor (NCoR/NCOR1), SMRT, repressor element 1-silencing transcription corepressor 1 (RCOR1/CoREST), and ligand-dependent nuclear receptor corepressor (LCoR). CoREST functions as a scaffold protein that associates with several histone-modifying enzymes, including lysine-specific demethylase 1 (LSD1), euchromatic histone methyltransferase 1 (GLP) and 2 (G9a), C-terminal binding protein 1 (CtBP1) as well as histone deacetylases HDAC1 and HDAC2 (*Shi et al., 2003, 2004*). Knockdown of CoREST blocked suppression of *IL-6* by both E2 and resveratrol (*Figure 5D*), demonstrating that CoREST is required for ERα-mediated repression of *IL-6*. In contrast, knockdown of LSD1, HDAC1, HDAC2, HDAC3, G9a, GLP, and several other ERα-interacting corepressors

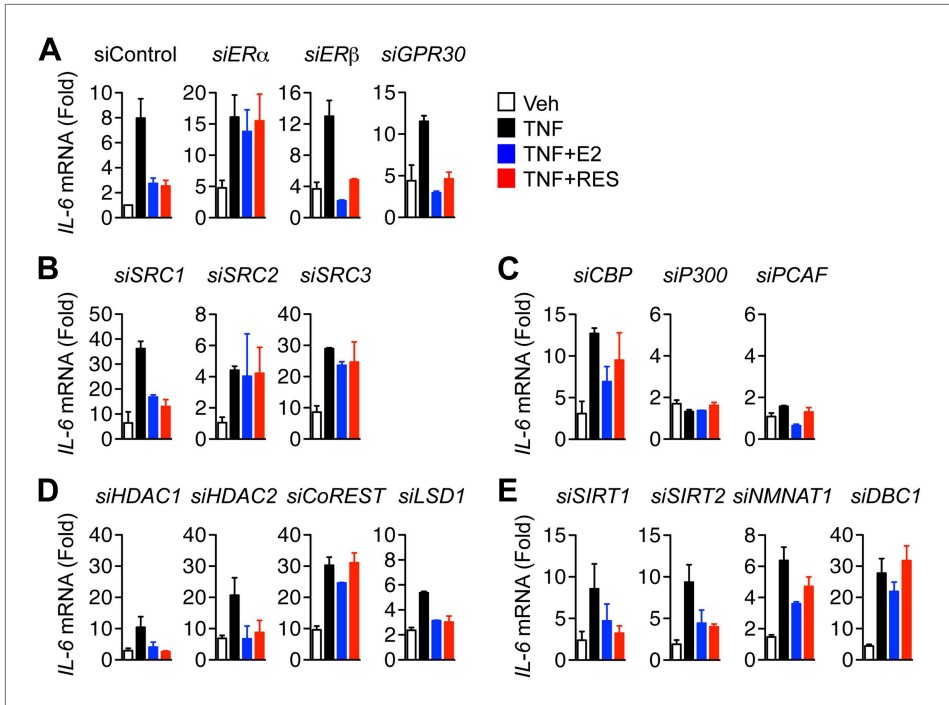

**Figure 5**. Molecular requirements for resveratrol- and E2-mediated suppression of *IL-6*. (**A–E**) MCF-7 cells were transfected with the indicated siRNAs and steroid-deprived for 48 hr. The cells were then treated with 10 ng/ml TNFα and 10 µM resveratrol or 10 nM E2 for 2 hr. *IL-6* mRNA levels were compared by qPCR, and are shown relative to the cells treated with ethanol vehicle and control siRNA. The small-scale siRNA screen was repeated three different times. Mean ± SEM are shown.

The following figure supplements are available for figure 5:

**Figure supplement 1**. Molecular requirements for resveratrol- and E2-mediated suppression of *IL-6*.

**Figure supplement 2**. Effect of siRNAs on target mRNA levels.

including SMRT, NCoR, LCoR, and CtBP1 (*Fernandes et al., 2003*; *Garcia-Bassets et al., 2007*), had no effect on suppression of *IL-6* by either E2 or resveratrol (*Figure 5D*, *Figure 5—figure supplement 1*). However, knockdown of HDAC2 siRNA globally raises expression of *IL-6*, as did knockdown of CoREST. Collectively, these findings suggest that CoREST is a dedicated corepressor required for ERα-mediated transrepression, but that it also has a more general role in limiting *IL-6* expression.

Knockdown of SIRT1 or SIRT2 had little or no effect on the suppression of *IL-6* by the ER ligands (*Figure 5E*). Indeed, SIRT1 siRNA slightly raised the expression of *IL-6* in cells treated with vehicle but not TNFα. However, two other proteins known to associate with SIRT1, nicotinamide mononucleotide adenylyltransferase 1 (NMNAT1) and deleted in breast cancer 1 (DBC1) (*Zhang et al., 2009*; *Yu et al., 2011a*), contributed to ligand-dependent repression of *IL-6* (*Figure 5E*). In contrast, depletion of poly ADP-ribose polymerase 1 (PARP1), which also interacts with SIRT1 (*Rajamohan et al., 2009*; *Zhang et al., 2012*), had no obvious effect on ERα-mediated repression of *IL-6* (*Figure 5—figure supplement 1*). Thus ERα requires a distinct, functionally diverse cohort of coregulators to mediate ligand-dependent transrepression of *IL-6*.

## Resveratrol-mediated inhibition of *IL-6* is independent of the PDE/cAMP and AMPK pathways

Although knockdown studies suggested that SIRT1 is not required for resveratrol-mediated suppression of *IL-6*, resveratrol is best known as a SIRT1 activator. Further, the lack of phenotype in a screening mode could reflect a number of alternatives for any of the individual siRNAs, including functional redundancies, lack of sufficient knockdown, and slow protein turnover. Consequently, we wanted to

assess the contribution of other resveratrol signaling pathways to resveratrol-dependent repression of *IL-6* (*Figure 6A*). Recently, some of these effects were shown to occur via resveratrol binding to and inhibiting cAMP-specific phosphodiesterases (PDEs) that hydrolyze and deplete cAMP (*Park et al., 2012*; *Tennen et al., 2012*). Thus, resveratrol elevates cellular cAMP levels and stimulates the canonical cAMP-signaling network downstream of catecholamine and glucagon signals that activate protein kinase A (PKA) and the CREB transcription factor, as well as rapid AMP-activated protein kinase (AMPK) signaling. In turn, AMPK drives production of NAD$^+$, the cofactor required for SIRT1 deacetylase activity, although signaling in the opposite direction has also been reported, where resveratrol stimulates SIRT1 via an unknown mechanism to activate AMPK (*Price et al., 2012*).

Resveratrol increased intracellular NAD$^+$ levels, and this increase was statistically significant at a resveratrol dose of 100 μM (*Figure 6B*). These findings suggest that the PDE/cAMP and AMPK pathways for NAD$^+$ production are active in this context, but at higher doses of resveratrol than required for anti-inflammatory effects through ERα. This raises the possibility that resveratrol represses *IL-6* via both ERα- and PDE-mediated mechanisms (*Figure 6A*). However the AMPK inhibitor, Dorsomorphin, did not affect repression of *IL-6* by E2 or resveratrol (*Figure 6—figure supplement 1*). Further, knockdown of both catalytic subunits of AMPK increased *IL-6* expression globally, but did not affect resveratrol-mediated repression of *IL-6* (*Figure 6C*, *Figure 6—figure supplement 2*). Finally, activation of the cAMP pathway with forskolin, or the PDE inhibitor rolipram, increased *IL-6* expression (*Figure 6D*), demonstrating categorically that activation of this pathway does not inhibit *IL-6* expression.

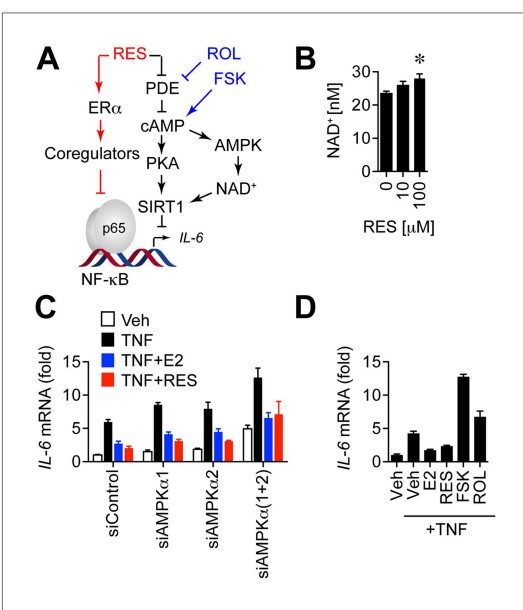

**Figure 6**. Resveratrol does not repress IL-6 through the cAMP or AMPK pathways. (**A**) Resveratrol stimulates ERα activity and inhibits cAMP-specific phosphodiesterases (PDEs) to activate cAMP SIRT1, and AMPK. The small molecule compounds i.e., the adenylyl cyclase activator forskolin (FSK) and the PDE inhibitor rolipram (ROL) used to further dissect this signaling network are shown in blue. (**B**) Resveratrol increases intracellular NAD$^+$ levels. Average intracellular NAD$^+$ concentrations were determined in MCF-7 cells treated with resveratrol for 5 min. Unpaired Student's *t* test (mean ± SEM, *n* = 6) was used to determine statistical significance. *p=0.006. (**C**) *IL-6* mRNA levels in steroid-deprived MCF-7 cells transfected with the indicated siRNAs and treated as described in *Figure 5*, were compared by qPCR. Mean ± SEM of representative biological duplicates are shown. (**D**) Steroid-deprived MCF-7 cells were treated with 10 ng/ml TNFα, 10 nM E2, 10 μM resveratrol, 10 μM FSK, and 25 μM ROL as indicated for 2 hr. Relative *IL-6* mRNA levels were compared by qPCR. Mean ± SEM of representative biological duplicates are shown.

The following figure supplements are available for figure 6:

**Figure supplement 1**. Resveratrol represses *IL-6* in cells dorsomorphin-treated cells.

**Figure supplement 2**. Effect of siRNAs on the mRNA levels of AMPK catalytic subunits.

## Resveratrol triggers ERα-mediated coregulator exchange at the *IL-6* promoter

To further probe the mechanism of ERα-mediated transrepression, chromatin immunoprecipitation (ChIP) assays were used to compare protein recruitment and accumulation of PTMs at the *IL-6* promoter. TNFα led to recruitment of ERα and the p65 NF-κB subunit, which were unaffected by E2 or resveratrol (*Figure 7A*). As a control, ChIP using pre-immune rabbit IgG showed no changes in promoter occupancy (*Figure 7—figure supplement 1*). In addition, resveratrol alone did not induce recruitment of ERα to *IL-6* promoter (*Figure 7—figure supplement 2*), as we have previously reported for the effects of E2 on several inflammatory genes (*Nettles et al., 2008b*). TNFα also led to accumulation of p65 acetylated at Lys310 (p65 K310-ac), a PTM catalyzed by p300 that is essential for full transcriptional activity (*Chen and Greene, 2004*), while resveratrol and E2 reduced p65 K310-ac levels (*Figure 7A*).

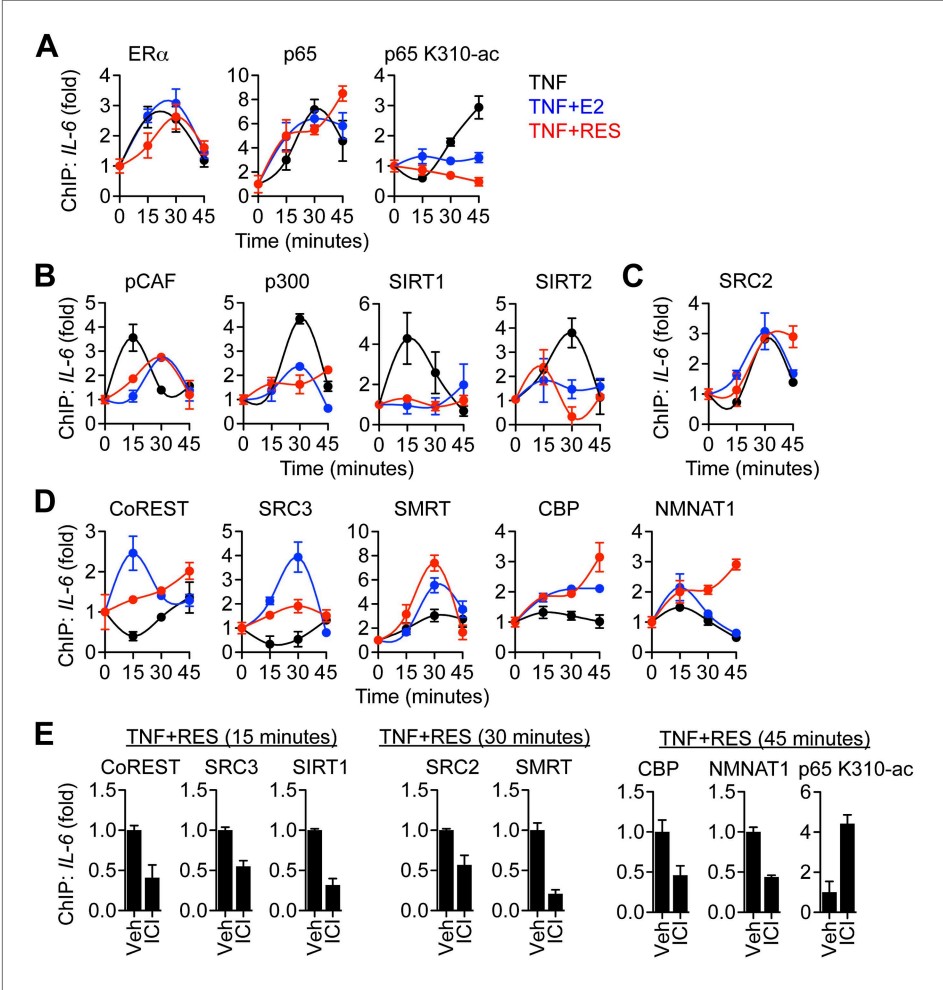

**Figure 7**. ERα orchestrates ligand-dependent coregulator exchange at the *IL-6* promoter. (**A–D**) Occupancy of the indicated factors at the *IL-6* promoter were compared by ChIP assay in steroid-deprived MCF-7 cells treated with 10 ng/ml TNFα alone or in combination with 10 nM E2 or 10 μM resveratrol, and fixed after 0, 15, 30, and 45 min (mean ± SEM *n* = 3) (**E**) Effect of ICI on promoter occupancy was determined by ChIP assay in steroid-deprived MCF-7 cells were pretreated with vehicle or 1 μM ICI for 1 hr, stimulated with 10 ng/ml TNFα plus 10 μM resveratrol, and fixed after 15, 30, or 45 min. Average promoter occupancies are shown as fold changes (mean ± SEM *n* = 3).

The following figure supplements are available for figure 7:

**Figure supplement 1**. Control ChIP assay.

**Figure supplement 2**. Without TNFα, RES does not induce recruitment of ERα to the *IL-6* promoter.

**Figure supplement 3**. RES induces ERα and SIRT1 recruitment at the *TFF1/pS2* promoter.

**Figure supplement 4**. ICI increased p65 K310-ac levels at the *IL-6* promoter.

**Figure supplement 5**. ICI did not increase recruitment of pCAF, p300 and SIRT2.

Resveratrol and E2 also reduced the recruitment of several coregulators. TNFα led to recruitment of pCAF, followed by p300, while E2 and resveratrol delayed recruitment of pCAF, and inhibited recruitment of p300 (*Figure 7B*), consistent in their ability to reduce levels of p65 K310-ac levels and *IL-6* expression. TNFα signaling also led to recruitment of SIRT1, followed by SIRT2, and resveratrol and E2 inhibited recruitment of both sirtuins at the *IL-6* promoter (*Figure 7B*). However, this was not

the case at the estrogen-induced *pS2* promoter, where resveratrol and E2-induced ERα and SIRT1 recruitment (*Figure 7—figure supplement 3*). It is noteworthy that SRC2 was required for coactivation by TNFα and for suppression of *IL-6* by ER ligands, but its recruitment was similar across signals (*Figure 7C*).

Resveratrol and E2 also modulated the recruitment of coregulators that showed some ligand-dependent differences. TNFα evicted CoREST and SRC3, whereas E2 and resveratrol led to recruitment of both factors to the *IL-6* promoter (*Figure 7D*). Resveratrol induced less recruitment of CoREST and SRC3 than E2, consistent with the reduced recruitment of SRC3 by resveratrol-bound ERα in the context of full-length proteins (*Figure 1D,F*). Resveratrol and E2 also augmented recruitment of SMRT, CBP and NMNAT1, and the effects of resveratrol were slightly greater than E2 (*Figure 7D*).

To determine if ERα mediated these events at the *IL-6* promoter, ChIP assays were performed in MCF-7 cells pre-treated with vehicle or the ER antagonist, ICI, and then treated with TNFα and resveratrol for an interval that showed a maximal effect, as determined from *Figure 5A–D*. ICI reduced recruitment of key coregulators, including CoREST, SRC3, CBP and NMNAT1 (*Figure 7E*), as well as coregulators such as SMRT and SIRT1 that were not required for resveratrol-dependent suppression of *IL-6*. ICI also increased p65 K310-ac levels (*Figure 7E*, *Figure 7—figure supplement 4*), consistent with higher NF-κB activity. It is interesting that ICI did not stimulate recruitment of pCAF, p300, or SIRT2 (*Figure 7—figure supplement 5*), suggesting that ICI can block ligand-induced activity, but does not mimic the unliganded receptor. Overall, the data demonstrate that resveratrol mediates repression of *IL-6* by orchestrating an ERα- and ligand-dependent exchange of a number of distinct coregulators that are required for the integration of steroidal and inflammatory signaling pathways (*Figure 8*).

## Discussion

### Pathway-selective ERα signaling

We demonstrate that resveratrol is a pathway-selective ERα ligand that modulates the inflammatory response without stimulating proliferation, by binding dynamically to the receptor, inducing an altered AF2 coactivator-binding site, and regulating the recruitment of a cast of coregulators at the *IL-6* locus. There is a large body of literature on resveratrol-mediated suppression of *IL-6*, as part of the inflammatory response in a variety of tissues, including liver, microglia, gut, and cardiovascular system, which are all ERα-positive tissues (*Csiszar et al., 2008*; *Pfluger et al., 2008*; *Lu et al., 2010*; *Singh et al.,*

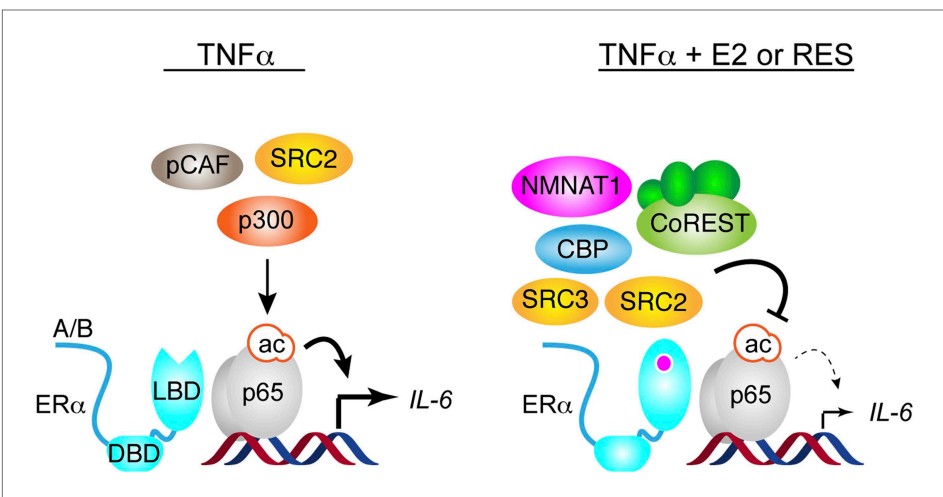

**Figure 8**. Proposed model for ERα-mediated transrepression of *IL-6*. In MCF-7 cells stimulated with TNFα, the p65 NF-κB subunit binds the *IL-6* promoter and mediates recruitment of many coregulators including p300, which acetylates p65 at Lys310, to drive transactivation of *IL-6*. In these cells, TNFα also induces recruitment of ERα to this site via a tethering mechanism. In response to E2 or resveratrol, ERα undergoes a conformational change, dismisses the set of coregulators including p300, and recruits a set that contains SRC3, CoREST, and other key coregulators required to inhibit p65 acetylation and repress *IL-6*.

*2010*). There is also evidence that the in vivo effects of resveratrol on the inflammatory response require ERs (*Yu et al., 2008*), but through previously unknown mechanisms. In this study, we show that the effects of resveratrol are ERα-dependent, and that resveratrol alters recruitment of the coregulators associated with ERα, thereby establishing ERα as the primary target for resveratrol modulation of the inflammatory response.

Our results support the concept that subtle modulation of receptor–coregulator interactions is sufficient to drive highly divergent phenotypes. This is shown by the reduced interaction of resveratrol-bound ERα with SRC3 in a mammalian two-hybrid assay, and reduced ERα-mediated recruitment of SRC3 to both the estrogen-stimulated gene, *GREB1*, and the estrogen-repressed gene, *IL-6*. While the original report of resveratrol as an ERα ligand described it as a superagonist (*Gehm et al., 1997*), many subsequent reports have described it as a partial agonist, and non-proliferative in the breast and uterus (*Ashby et al., 1999*; *Turner et al., 1999*; *Bowers et al., 2000*; *Bhat et al., 2001*; *Xu and Li, 2003*; *Karmakar et al., 2009*). Further, there have been a number of clinical trials of resveratrol in humans, without reports of feminization (*Tome-Carneiro et al., 2013*). We found that pathway-selective resveratrol action was associated with changes in the AF2 surface of the LBD, but not differences in affinity between the short LxxLL motif peptides derived from different members of the SRC family. Instead, the determinants of SRC-binding selectivity may be just C-terminal to the ordered part of the receptor-interaction domain (*Scheinman et al., 1995*), may lie further outside the SRC regions tested (*Leo and Chen, 2000*), and might involve the other functional domains of ERα outside the LBD. In fact, SRC2 also interacts with ERα via the AF1 coactivator-binding site located in the unstructured N-terminus of the receptor (*Norris et al., 1998*). The peptide profiling experiments show that resveratrol generally lowers affinity for recruited peptides, but display a defect in dismissal of peptides bound to the unliganded LBD, thus demonstrating a change in the shape of the AF2 surface in solution. Also, functional analysis of ERα domains suggests that the DNA-binding domain plays a vital role in resveratrol-induced ERα activity (*Srinivasan et al., 2013*). These data support the idea that inter-domain communication and binding of coactivators to multiple ERα domains is an important aspect of this anti-inflammatory-selective signaling mechanism.

Resveratrol belongs to a newly discovered class of compounds that can bind to ERα in two different orientations. With either the phenol or the resorcinol group forming the conserved hydrogen bond with helix 3, the ensemble of receptors will display a mixture of conformers, including potentially dimers with different combinations of binding modes. Importantly, we previously showed that binding of ligands in two flipped orientations could stabilize the receptor in either the active or inactive conformations, generating partial agonist activity (*Bruning et al., 2010*). Further, those compounds could be modified to titrate the relative balance of stabilizing the active vs inactive protein conformations.

Ligand dynamics as an allosteric control mechanism represents a new principle in drug design that has since been observed with PPARγ (*Hughes et al., 2012*), dihydrofolate reductase (*Carroll et al., 2011*), and more recently a mechanism to generate partial agonists for G protein-coupled receptors (*Bock et al., 2014*). In addition, we found that dynamic binding of ligands also contributes to pathway-selective signaling, which like resveratrol, was selectively anti-inflammatory (*Srinivasan et al., 2013*). Thus, the multiple binding modes for resveratrol may contribute to its reduced gene activation signal and lack of a proliferative effect.

## Mechanisms of signal integration

Signaling from estrogens or pro-inflammatory cues involves spatio-temporal coordination of complex transcriptional activation programs (*Shang et al., 2000*; *Metivier et al., 2003*; *Medzhitov and Horng, 2009*). Kinetic ChIP assays at a single locus are an important addition to genome-scale ChIP studies, and they have revealed that signal integration can involve shifts in the timing of chromatin association. For example, pCAF recruitment to the *IL-6* promoter is dynamically regulated in a distinct fashion by different signaling cues. Likewise, our results suggest that estrogen- and resveratrol-dependent attenuation of the inflammatory response is not simply a blockade of a single signaling pathway, but requires ERα-mediated orchestration of complex transcriptional repression programs. At the *IL-6* promoter, one aspect of this repression program involves recruitment of SRC3 and CBP, ligand-dependent dismissal of p300, and loss of p65 K310-ac, which is required for full transcriptional activity and which could be directed by p300 (*Chen and Greene, 2004*). Our results support a model where resveratrol-bound ERα mediates recruitment of an

SRC3/CBP complex and blocks the TNFα-induced recruitment of p300 and pCAF, thereby blocking acetylation of p65 (*Figure 8*).

The initial description of coregulators as either coactivators or corepressors has evolved with the understanding that they have more context-specific effects. The opposing effects of CBP and p300, and the different roles of SRC2—coactivating TNFα induction of *IL-6*, but corepressing ERα-mediated signaling on the same gene—support this idea. The disparate roles of SRCs are also striking and unexpected, as all three played some role in repressing *IL-6*. SRC1 and SRC3 played ligand-independent roles, while SRC2 and SRC3 were more specifically required for repression by E2 and resveratrol. These differences are likely due to the different transient, multi-protein complexes formed by these promiscuous coregulators (*Stenoien et al., 2001*; *Jung et al., 2005*; *Malovannaya et al., 2011*). For example, the mouse ortholog of SRC2, called GRIP1, was found to have an additional role in glucocorticoid-mediated repression of inflammatory genes (*Rogatsky et al., 2002*), which mapped to a binding site for a trimethyltransferase, Suv4-20h1, an enzyme that represses glucocorticoid receptor activity (*Chinenov et al., 2008*). A similar context-dependent activity is also seen with the corepressor, SMRT, which is required for activation of some ERα-target genes (*Peterson et al., 2007*). The preferred association of resveratrol-bound ERα with SRC2 is also intriguing, given roles of both resveratrol and SRC2 in metabolic regulation (*York and O'Malley, 2010*). Interestingly, recruitment of p65 and ERα were largely insensitive to E2 and resveratrol, suggesting that these ligands change the conformation of ERα at the promoter to dictate the shape of the AF2 surface and modulate recruitment of SRCs and other coregulators, but also to change the structure of proteins such as SRC2, which shows changes in function despite similar recruitment profiles (*Figure 8*).

Other coregulators, such as the scaffold CoREST, form biochemically stable complexes (*Shi et al., 2005*; *Malovannaya et al., 2011*), which may provide a less flexible platform for signal integration, but which brings together a dedicated group of effector enzymes. The lack of phenotypes from targeting the enzyme components of the CoREST complex does not necessarily indicate that these targets are not involved in ERα-mediated transrepression, as the siRNA screen showed variable knockdown, and target-specific optimizations might be required to reveal their effects. Moreover, this may also reflect functional redundancy, for example of HDAC1 and HDAC2, or G9a and GLP. However, HDAC2 siRNA increased basal expression of *IL-6*, suggesting that these subunits are required to restrain *IL-6* expression in a TNFα- and ERα-independent manner, consistent with previous ChIP-array studies in MCF-7 cells, which suggest that *IL-6* is a target of an LSD1/CoREST/HDAC complex (*Wang et al., 2009*). The ability to perturb and track many coregulators in parallel illustrates that multiple determinants contribute to a single phenotype such as *IL-6* expression, similar to the different coregulator requirements of estrogen-induced genes (*Won Jeong et al., 2012*).

## Polypharmacology of resveratrol

While knockdown of SIRT1 had no major effect on *IL-6* expression in breast cancer cells, ERα-driven control of the association of SIRT1 with chromatin contributes to SIRT1 activity in other contexts (*Elangovan et al., 2011*; *Yu et al., 2011a*). Indeed, we show here that ERα ligands can direct SIRT1 to a canonical ERE of an estrogen-induced gene, *pS2*, while blocking TNFα-induced recruitment of SIRT1 to the *IL-6* promoter. Further, several approaches established that activation of the cAMP or AMPK pathways were not required for resveratrol-directed suppression of *IL-6*, and in fact, forskolin strongly induced *IL-6* expression. Thus, resveratrol regulates SIRT1 through several possible mechanisms, including via ERα, as established here.

This polypharmacology likely accounts for the unique health benefits of resveratrol in different preclinical models. For example, in the muscle the beneficial metabolic effects of resveratrol may be via ERα-directed induction of Glut4 and increased glucose uptake (*Deng et al., 2008*), up-regulation of cAMP signaling (*Park et al., 2012*), PGC-1α expression (*Pfluger et al., 2008*), mitochondrial biogenesis (*Price et al., 2012*), and activation of the AMPK (*Patel et al., 2011*; *Price et al., 2012*) or PPARγ (*Ge et al., 2007*). Thus, dissecting the effects of resveratrol requires consideration of several potential signaling pathways, as well as tissue context. This work advances our understanding of resveratrol, which acts through ERα to modulate the inflammatory response, without the proliferative effects of estradiol. Therefore, this work will impact future medicinal chemistry efforts to improve the potency or efficacy of resveratrol.

## Materials and methods

### Cell culture

MCF-7 and T47D cells were cultured in growth medium containing Dulbecco's minimum essential medium (DMEM) (Cellgro by Mediatech Inc, Manassas, VA) plus 10% fetal bovine serum (FBS) (Hyclone by Thermo Scientific, South Logan, UT), and 1% each of nonessential amino acids (NEAA) (Cellgro), Glutamax and Penicillin-streptomycin-neomycin (PSN) antibiotics mixture (Gibco by Invitrogen Corp. Carlsbad, CA) and maintained at 37°C and 5% $CO_2$. For each experiment, MCF-7 cells are seeded in growth medium for 24 hr. The medium was then replaced with steroid-free medium containing phenol red-free DMEM plus 10% charcoal/dextran-stripped (cs) FBS, and 1% each of NEAA, Glutamax and PSN, and the cells were incubated at 37°C for 48–72 hr before treatment. The cells were pre-treated with 1 µM ICI 182,780 (ICI) or 1 µM in solution AMPK inhibitor compound C/Dorsomorphin (DOS) (Calbiochem, EMD Millipore Corp. Billerica, MA). The cells were treated simultaneously with the following, unless otherwise indicated: 10 ng/ml human tumor necrosis factor alpha (TNFα; Invitrogen), and 10 nM E2, 10 µM resveratrol (RES), 25 µM Rolipram (ROL), or 10 µM Forskolin (FSK) (Sigma–Aldrich Inc., St. Louis, MO).

### Luciferase assay

MCF-7 cells were transfected with a widely used 3xERE-luciferase reporter and luciferase activity was measured as previously described (*Wang et al., 2012*).

### Cell proliferation assay

MCF-7 cells were placed in 384-well plates containing phenol red-free growth media supplemented with 5% charcoal-dextran sulfate-stripped FBS, and stimulated with ER ligands the next day, using a 100 nl pintool Biomeck NXP workstation (Beckman Coulter, Inc.). After 3 days, the treatments were repeated. The number of cells/well was determined using CellTitre-Glo reagent (Promega Corp., Madison, WI) as previously described (*Srinivasan et al., 2013*), 7 days after the initial treatment.

### Mammalian two-hybrid assay

HEK293-T cells were transfected with ERα-VP16 and either GAL4-SRC1, GAL4-SRC2 or GAL4-SRC3 and GAL4-UAS-Luciferase using TransIT LT1 transfection reagent (Mirus Bio LLC, Madison, WI), processed and analyzed as previously described (*Srinivasan et al., 2013*).

### IL-6 ELISA

Aliquots of media conditioned by stimulated MCF-7 and RAW264.7 macrophages were respectively analyzed using human IL-6 or mouse Il-6 AlphaLISA no-wash ELISA kits (PerkinElmer, Inc., Shelton, CT), as previously described (*Srinivasan et al., 2013*).

### Gene expression analyses

Total RNA was isolated using the RNeasy Mini kit (Qiagen Inc., Valencia, CA) and submitted to the Scripps-Florida genomics core for cDNA microarray analysis using Affymetrix Genechip Human Gene ST arrays. For high-throughput, quantitative RT-PCR (qPCR), 1 µg of total RNA per sample was reverse-transcribed in a 20-µl reaction using a High capacity cDNA kit (Applied Biosystems, Carlsbad, CA). 1 µl of the resulting cDNA mixture was amplified in a 10 µl reaction using gene-specific primers (*Tables 2 and 3*) in 1x Taqman or SYBR green PCR mixes (Applied Biosystems). Data were analyzed using the ΔΔCT method as previously described (*Bookout and Mangelsdorf, 2003*) and GAPDH expression as an endogenous control (Product number: 4333764F; Applied Biosystems).

### X-ray crystallography

The ERα ligand-binding domain containing an Y537S mutation was expressed in *E. coli* and purified as previously described (*Nettles et al., 2008a*). The protein solution was mixed with resveratrol and a receptor-interacting SRC2 peptide, and allowed to crystallize at room temperature. X-ray diffraction data on the crystal was collected at Stanford Synchrotron Radiation Lightsource beam line 11-1. The structure was solved via automated molecular replacement and rebuilding of the genistein-bound ERα (PDB 2QA8) (*Nettles et al., 2008a*), using the PHENIX software suite (*Adams et al., 2010*). Ligand docking was followed by series of ExCoR and rebuilding as previously described (*Nwachukwu et al., 2013*).

## Synthesis of F-resveratrol

i. P(OEt)$_3$, DMF, 160°C, 4h
ii. 3-fluoro-4-methoxybenzaldehyde, t-BuOK, DMF, 0°C-RT
iii. BBr$_3$, CH$_2$Cl$_2$, 0°C-RT, 20h

## Diethyl 3,5-dimethoxybenzylphosphonate S1

Triethylphosphite (750 µl, 4.3 mmol) and 3,5-dimethoxybenzaldehyde (0.981 g, 4.3 mmol) were sealed together in a pressure vial. The reaction was stirred while heating to 160°C for 4 hr. After being cooled to room temperature, it was concentrated under reduced pressure to yield 1.270 g (99% yield) of a clear oil **S1**. $^1$H NMR (499 MHz, CDCl$_3$) δ 6.44 (t, J = 2.4 Hz, 2H), 6.37–6.27 (m, 1H), 4.10–3.94 (m, 4H),

**Table 2.** Gene-specific qPCR primers

| Gene | Forward (5′-3′) | Reverse (5′-3′) |
|---|---|---|
| CTBP1 | CTCAATGGGGCTGCCTATAG | GGACGATACCTTCCACAGCA |
| DBC1 | GATCCACACACTGGAGCTGA | TGGCTGAGAAACGGTTATGG |
| G9a | CTTCAGTTCCCGAGACATCC | CGCCATAGTCAAACCCTAGC |
| GLP | GCTCGGGTTTGACTATGGAG | CAGCTGAAGAGCTTGCCTTT |
| GPR30 | CTGACCAAGGAGGCTTCCAG | CTCTCTGGGTACCTGGGTTG |
| HDAC1 | AAGGAGGAGAAGCCAGAAGC | GAGCTGGAGAGGTCCATTCA |
| HDAC2 | TCCAAGGACAACAGTGGTGA | GTCAAATTCAGGGGGTTGCTG |
| HDAC3 | AGAGGGGTCCTGAGGAGAAC | GAACTCATTGGGTGCCTCTG |
| LCoR | CTCTCCAGGCTGCTCCAGTA | ACCACTCCGAAGTCCGTCT |
| LRP16 | AGCACAAGGACAAGGTGGAC | CTCCGGTAGATGTCCTCGTC |
| LSD1 | GGCTCAGCCAATCACTCCT | ATGTTCTCCCGCAAAGAAGA |
| ROCK1 | CCACTGCAAATCAGTCTTTCC | ATTCCACAGGGCACTCAGTC |
| SIRT2 | TTGGATGGAAGAAGGAGCTG | CATCTATGCTGGCGTGCTC |
| SRC1 | CACACAGGCCTCTACTGCAA | TCAGCAAACACCTGAACCTG |
| SRC2 | AGCTGCCTGGAATGGATATG | AACTGGCTTCAGCAGTGTCA |
| SRC3 | GTGGTCACATGGGACAGATG | TCTGATCAGGACCCATAGGC |

**Table 3.** Inventoried TaqMan gene expression assays (Applied Biosystems)

| Gene | Assay ID |
| --- | --- |
| AMPKα1 | Hs01562315_m1 |
| AMPKα2 | Hs00178903_m1 |
| CBP | Hs00231733_m1 |
| CoREST | Hs00209493_m1 |
| ERα | Hs01046812_m1 |
| ERβ | Hs01100353_m1 |
| IL-6 | Hs00174131_m1 |
| NCoR | Hs01094540_m1 |
| NMNAT1 | Hs00978912_m1 |
| P300 | Hs00914223_m1 |
| PARP1 | Hs00242302_m1 |
| PCAF | Hs00187332_m1 |
| SIRT1 | Hs01009006_m1 |
| SMRT | Hs00196955_m1 |

3.76 (d, J = 2.2 Hz, 6H), 3.07 (d, J = 21.7 Hz, 2H), 1.29–1.16 (m, 6H). $^{13}$C NMR (126 MHz, CDCl$_3$) δ 160.6, 133.6, 107.7, 99.0, 62.0, 55.2, 34.4, 16.3. HRMS (ESI$^+$) m/z calculated for $C_{13}H_{32}O_5P^+$289.1205, found 289.1197. See *Figure 9*.

## (*E*)-1-(3-fluoro-4-methoxystyryl)-3,5-dimethoxybenzene S2

Under nitrogen, an oven-dried flask was charged with diethyl 3,5-dimethoxybenzylphosphonate (292 mg, 1 mmol), which was dissolved in anhydrous dimethylformamide (2.5 ml) and cooled to 0°C. 3-Fluoro-4-methoxybenzaldehyde (166 mg, 1 mmol) was added. Potassium *tert*-butoxide (227 mg, 2 mmol, 2 eq.) was added, and the cloudy red mixture was stirred for 90 min, while being allowed to warm to room temperature. The reaction was quenched with water (15 ml). The organic products were extracted with ethyl acetate (2 × 25 ml), washed with brine (50 ml) and dried over anhydrous Na$_2$SO$_4$. After filtration, the crude material was concentrated under reduced pressure. The product was purified by column chromatography (silica gel with 10:1 hexane:ethyl acetate) to yield 0.209 g (72% yield) of a white solid **S2**. $^1$H NMR (500 MHz, CDCl$_3$) δ 7.26 (dd, J = 12.6, 2.1 Hz, 1H), 7.19–7.11 (m, 1H), 7.00–6.82 (m, 3H), 6.65 (d, J = 2.3 Hz, 2H), 6.41 (s, 1H), 3.88 (s, 3H), 3.82 (s, 6H). $^{13}$C NMR (126 MHz, CDCl$_3$) δ 160.8, 153.3, 151.3, 147.2, 147.1, 139.0, 130.5, 127.7, 127.5, 122.9, 113.3, 113.1, 104.3, 99.7, 56.0, 55.1. HRMS (ESI$^+$) m/z calculated for $C_{17}H_{18}O_3F^+$ 289.1240, found 289.1234. See *Figure 10*.

## (*E*)-5-(3-fluoro-4-hydroxystyryl)benzene-1,3-diol F-Resveratrol

Under nitrogen, (*E*)-1-(3-fluoro-4-methoxystyryl)-3,5-dimethoxybenzene (117 mg, 0.4 mmol) was suspended in anhydrous dichloromethane (1.6 ml). The reaction mixture was cooled to 0°C. A 1.0 M solution of boron tribromide in dichloromethane (4.0 ml, 4 mmol, 10 eq.) was added slowly dropwise over the course of 25 min. The reaction was stirred overnight and allowed to warm to room temperature. The reaction was quenched with saturated NaHCO$_3$ (20 ml). The organic products were extracted with ethyl acetate (3 × 50 ml), washed with water (50 ml), dried over anhydrous Na$_2$SO$_4$, filtered, and concentrated yielding a 76 mg (76% yield) of a brown solid **S3**. $^1$H NMR (500 MHz, CD$_3$OD) δ 7.23 (dd, J = 12.5, 2.1 Hz, 1H), 7.15–7.06 (m, 1H), 6.96–6.77 (m, 3H), 6.48 (d, J = 2.2 Hz, 2H), 6.21 (t, J = 2.2 Hz, 1H), 4.94 (bs, 3H). $^{13}$C NMR (126 MHz, CD$_3$OD) δ 159.5, 153.9, 152.0, 145.6, 145.5, 140.8, 131.3, 128.4, 128.3, 124.1, 124.0, 118.7, 114.3, 114.2, 105.9, 102.9. HRMS (ESI$^+$) m/z calculated for $C_{14}H_{12}O_3F^+$ 247.0770, found 247.0773. See *Figure 11*.

### F19-NMR

F-resveratrol was added to dilute ERα ligand binding domain (Y537S) in 15 ml of buffer (20 mM Tris pH 8.0, 150 mM NaCl, 5% glycerol, 15 mM BME), then concentrated, and 10% D$_2$O added for a final protein concentration of 260 µM with 0.2% DMSO-d6. $^{19}$F NMR was performed on a 700 MHz Bruker NMR spectrometer ($^{19}$F @ 659 MHz) without proton decoupling. Spectra were referenced to KF in buffer (set to 0 ppm) using a thin coaxial tube insertion.

### LanthaScreen

SRC peptide binding to the ERα ligand-binding domain (LBD) was examined using the LanthaScreen time-resolved fluorescence resonance energy transfer (FRET) ERα Coactivator Assay kit (Invitrogen Corporation, Carlsbad, CA), as previously described (*Choi et al., 2011*), but run in agonist mode. Specifically, 3.5 nM ERα-LBD-GST, 5 nM Terbium-tagged anti-GST antibody, fluorescein-tagged SRC peptides, and E2 or resveratrol were placed in triplicates in a 384-well plate, mixed, and incubated at room temperature for 1 hr in the dark. The FRET signals emitted upon excitation at 340 nm were read at 520 nm and 495 nm, and the emission ratio (520/495) from each well was calculated.

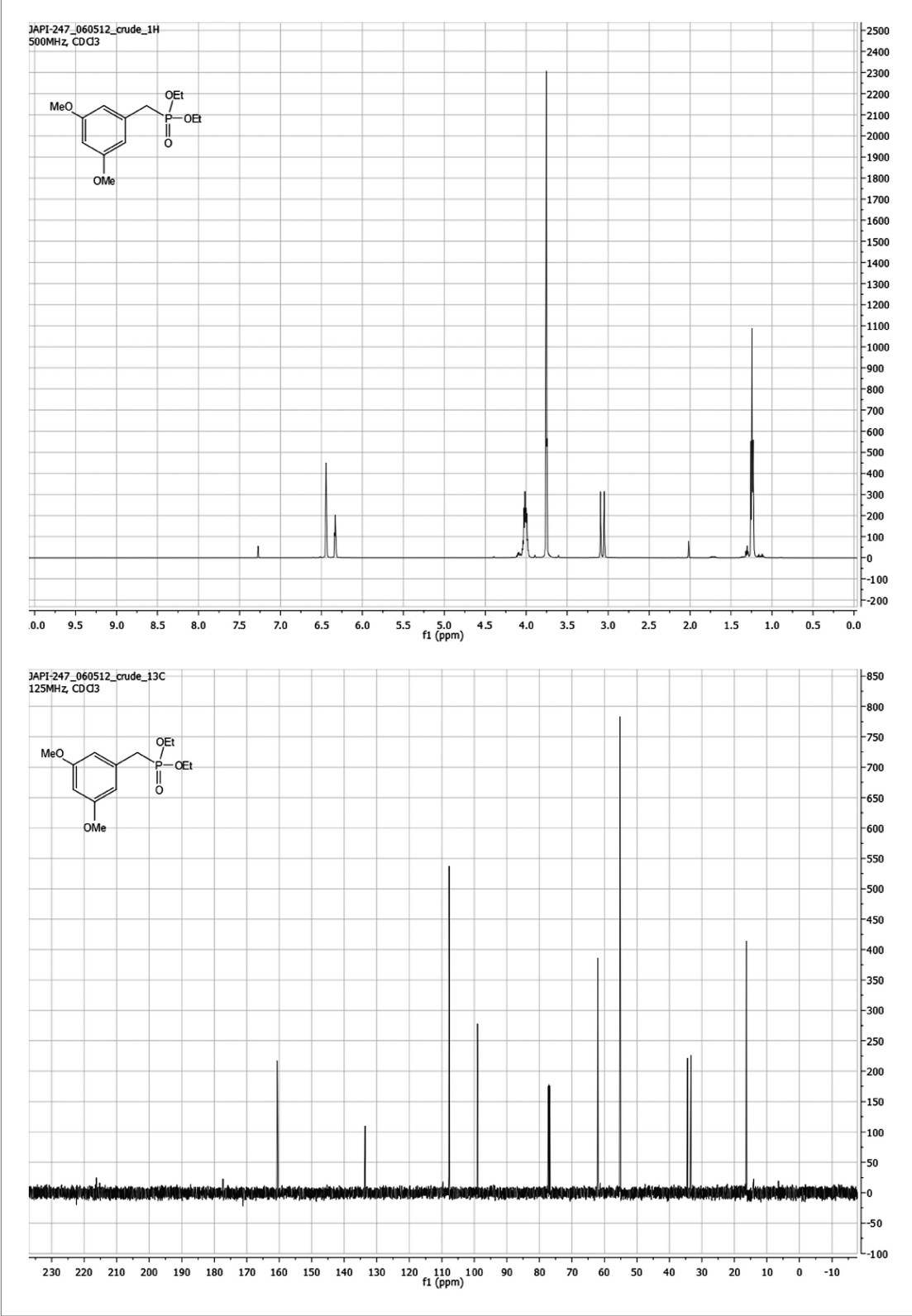

**Figure 9**. Diethyl 3,5-dimethoxybenzylphosphonate S1.

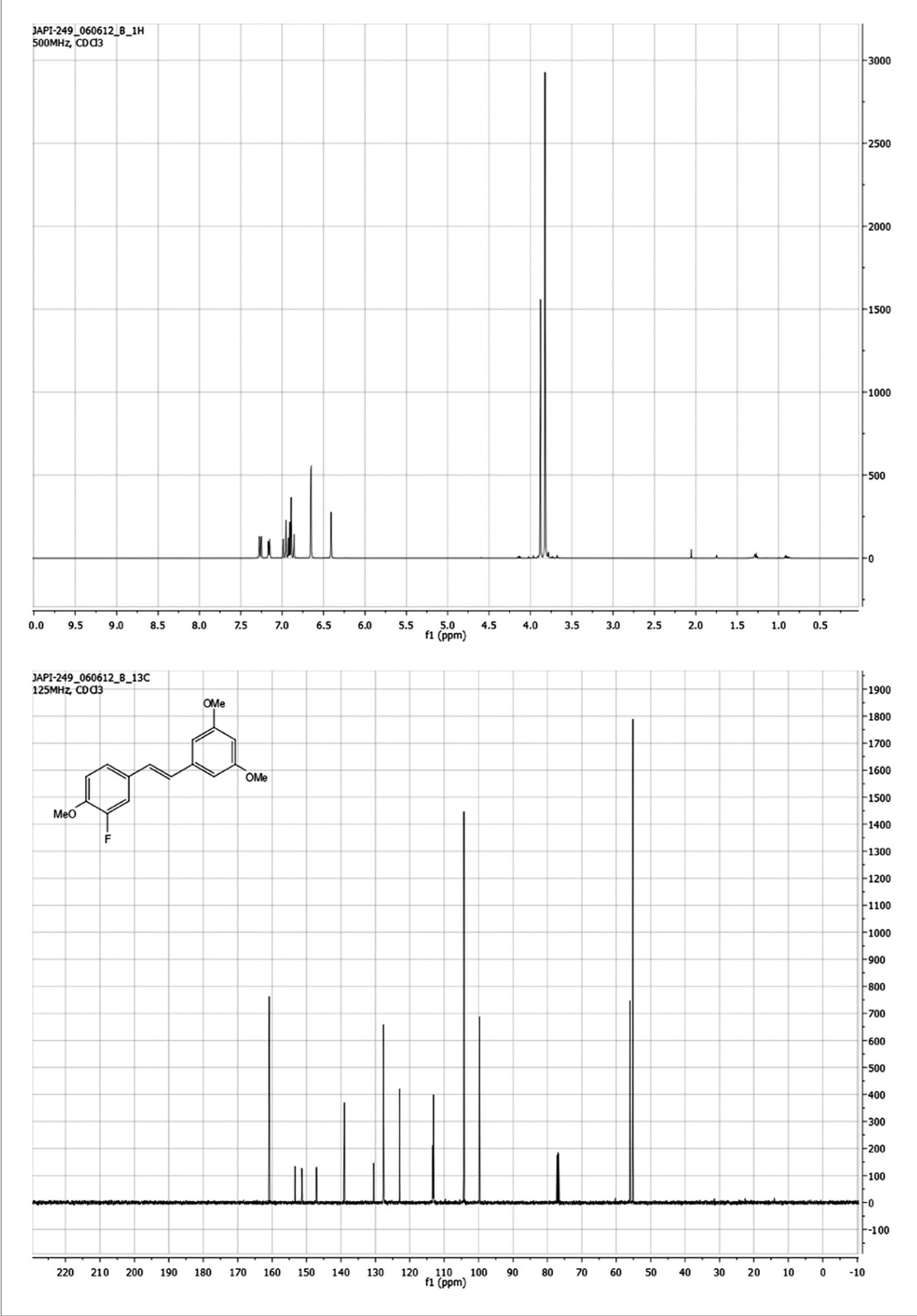

**Figure 10**. *(E)*-1-(3-fluoro-4-methoxystyryl)-3,5-dimethoxybenzene S2.

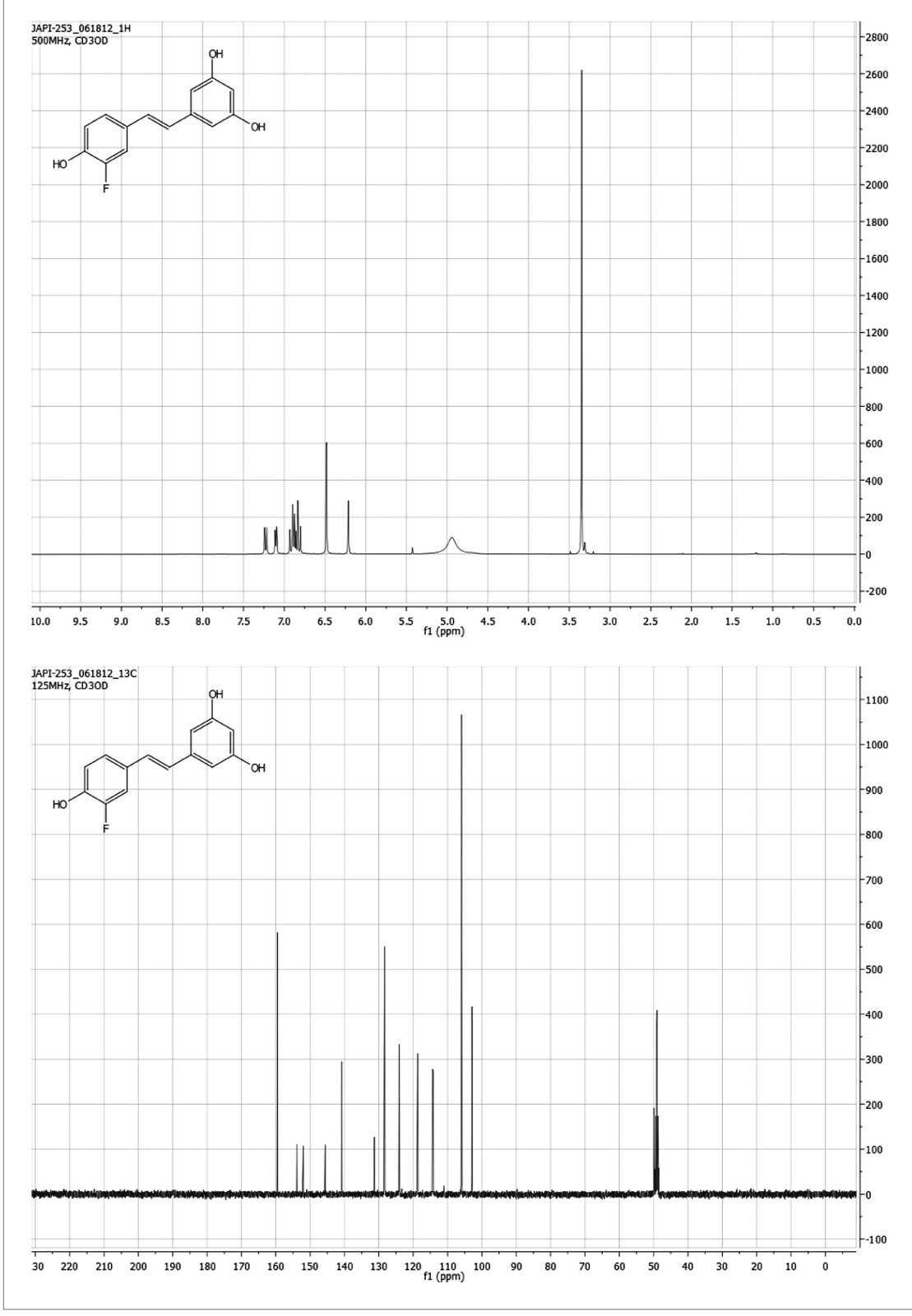

**Figure 11**. *(E)*-5-(3-fluoro-4-hydroxystyryl)benzene-1,3-diol F-Resveratrol.

## MARCoNI coregulator interaction profiling

Microarray assay for real-time nuclear receptor coregulator interaction (MARCoNI) was performed as previously described (*Aarts et al., 2013*). In short, a PamChip peptide micro array with 154 unique coregulator-derived NR interaction motifs (#88101; PamGene International) was incubated with His-tagged ERα LBD in the presence of 10 µM E2 or A-CD ring estrogen, 100 µM resveratrol, or solvent only (2% DMSO, apo). Receptor binding to each peptide on the array was detected using fluorescently labeled His-antibody, recorded by CCD and quantified. Per compound, three technical replicates (arrays) were analyzed to calculate the log-fold change (modulation index, MI) of each receptor–peptide interaction vs apo. Significance of this modulation was assessed by Student's *t* test.

## RNAi

MCF-7 cells were placed in a 24-well plate at a density of 50,000 cells/well for 24 hr. The next day, cells were transfected with 100 nM siRNAs (*Tables 4 and 5*) using X-tremeGENE siRNA transfection reagent (Roche Applied Science, Indianapolis, IN). For each well, a 25-µl mixture containing 2.5 µl X-tremeGENE + 22.5 µl Opti-MEM (Invitrogen) was added to a 25-µl solution of siRNA + Opti-MEM, mixed and incubated at room temperature for 20 min, and then added to cells in 0.45 ml Opti-MEM. After 6 hr, the media was replaced with steroid-free media and left for 48 hr before ligand stimulation.

AllStars negative control (siControl, Qiagen Inc.).

## NAD⁺ assay

MCF-7 cells were seeded in a 24-well plate at a density of 50,000 cells/well in growth medium for 24 hr. The medium was then replaced with steroid-free medium for 48 hr. The cells were stimulated with the indicated doses of resveratrol. After 5 min, the cells were washed with cold PBS, disrupted in 100 µl NAD extraction buffer, and analyzed using the EnzyChrom NAD⁺/NADH Assay kit (BioAssay Systems, Hayward, CA).

## High-throughput quantitative chromatin immuno-precipitation (ChIP) assay

MCF-7 cells in a 12- or 24-well plate were fixed, and washed with cold 1X PBS. 400 µl/well of cold lysis buffer was added to the cells which were then incubated at 4°C for 1 hr. Whole cell lysates were transferred to a 1.5-ml tube for sonication. For each IP, 100 µl aliquots of sonicated lysate was mixed with antibody and 25 µl Dynabeads protein G (Invitrogen) to make a 200 µl lysis buffer mixture that was rotated for 24 hr at 4°C. The precipitate was washed sequentially in previously described low salt, high salt, and LiCl buffers (*Nwachukwu et al., 2007*) and twice in 1x TE buffer, after which the crosslinks were reversed. DNA fragments were isolated using QIAquick PCR purification kit (Qiagen), and analyzed by qPCR using Taqman 2x PCR master mix and a custom FAM-labeled promoter probes (Applied Biosystems).

## 10 ml Of lysis buffer

26 mg Hepes
1 mM EDTA
0.5 mM EGTA
10 mM Tris–HCl pH 8.0
10% (vol/vol) Glycerol
0.5% (vol/vol) NP-40/Igepal CA630

**Table 4.** Flexitube siRNAs (Qiagen)

| siRNA | Gene ID | Catalog No. |
|---|---|---|
| AMPKα1 | 5562 | SI02622228 |
| AMPKα2 | 5563 | SI02758595 |
| CoREST | 23,186 | SI03137435 |
| CtBP1 | 1487 | SI03211201 |
| DBC1 | 57,805 | SI00461846 |
| GLP | 79,813 | SI02778923 |
| G9a | 10,919 | SI00091189 |
| ERα | 2099 | SI02781401 |
| ERβ | 2100 | SI03083269 |
| GPR30 | 2852 | SI00430360 |
| HDAC1 | 3065 | SI02663472 |
| HDAC2 | 3066 | SI00434952 |
| HDAC3 | 8841 | SI00057316 |
| LCOR | 84,458 | SI00143213 |
| LRP16 | 28,992 | SI00623658 |
| LSD1 | 23,028 | SI02780932 |
| NMNAT1 | 64,802 | SI04344382 |
| P300 | 2033 | SI02622592 |
| PARP1 | 142 | SI02662996 |
| SIRT1 | 23,411 | SI04954068 |
| SIRT2 | 22,933 | SI02655471 |
| SRC1 | 8648 | SI00055342 |
| SRC2 | 10,499 | SI00089509 |
| SRC3 | 8202 | SI00089369 |

**Table 5.** ON-TARGETplus SMARTpool siRNAs (Thermo Scientific Dharmacon, Lafayette, CO)

| siRNA | Gene ID | Catalog No. |
|-------|---------|-------------|
| PCAF  | 8850    | L-005055-00 |
| CBP   | 1387    | L-003477-00 |
| NCoR  | 9611    | L-003518-00 |
| SMRT  | 9612    | L-020145-00 |

0.25% (vol/vol) Triton-X 100
0.14 M NaCl
+ nuclease-free $H_2O$
+1x Protease Inhibitor cocktail (Roche)

## ChIP antibodies
CBP (A22), CoREST (E−15), ERα (HC-20), NMNAT1 (H-109), p300 (C-20), p65/RelA (C-20), pCAF (H-369), SMRTe (H-300), SRC2 (R-91), SRC3 (M-397), SIRT2 (A-5) and normal rabbit IgG (cat no. sc-2027) (Santa Cruz Biotechnology, Inc.).

SIRT1 (C14H4) (Cell Signaling Technology, Inc.).
Acetylated p65/RelA Lys310 (ab19870) (Abcam, Cambridge, MA).

## Custom TaqMan probe sequences
*GREB1* promoter (ERE 1)—Forward: 5′-GTGGCAACTGGGTCATTCTGA-3′; Reverse: 5′-CG ACCCACA GAAATGAAAAGG-3′; and FAM-probe: 5′-CGCAGCAGACAATGATGAAT-3′.

 *IL-6* promoter—Forward: 5′-CCCTCACCCTCCAACAAAGATTTAT-3′; Reverse: 5′-GCCTC AGACATC TCCAGTCCTATAT-3′; and FAM-probe: 5′-AAATGTGGGATTTTCC-3′.

*pS2/TFF1* promoter—Forward: 5′-CTAGACGGAATGGGCTTCATGAG-3′; Reverse: 5′-GCT TGGCCG TGACAACAG-3′; and FAM-probe: 5′-CCCCTGCAAGGTCACG-3′.

## Acknowledgements
We thank John Cleveland, Donald McDonnell, and Bert O'Malley for comments on the manuscript.

Research support from the National Institutes of Health (PHS 5R37DK015556 to JAK; 5R33CA132022, 5R01DK077085 to KWN; 1U01GM102148 to KWN and PRG; R01DK101871 to DJK; F30DK083899 to AAP; T32ES007326 to JAP; and F32DK097890 to TSH), the BallenIsles Men's Golf Association (to JCN), and the Frenchman's Creek Women for Cancer Research (to SS). DJK was also supported with start-up funds from The Scripps Research Institute and the James and Esther King Biomedical Research Program, Florida Department of Health (1KN-09).

Use of the Stanford Synchrotron Radiation Lightsource, SLAC National Accelerator Laboratory, is supported by the US Department of Energy, Office of Science, Office of Basic Energy Sciences under Contract No. DE-AC02-76SF00515. The SSRL Structural Molecular Biology Program is supported by the DOE Office of Biological and Environmental Research, and by the National Institutes of Health, National Institute of General Medical Sciences (including P41GM103393). The contents of this publication are solely the responsibility of the authors and do not necessarily represent the official views of NIGMS or NIH.

## Additional information

### Funding

| Funder | Grant reference number | Author |
|--------|------------------------|--------|
| National Institutes of Health | 5R37DK015556 | Jerome C Nwachukwu, Sathish Srinivasan, Jason Nowak, Kendall W Nettles, Alexander A Parent, Julie A Pollock, John A Katzenellenbogen |
| National Institutes of Health | 5R33CA132022 | Jerome C Nwachukwu, Sathish Srinivasan, Jason Nowak, Kendall W Nettles |
| National Institutes of Health | 1U01GM102148 | Jerome C Nwachukwu, Sathish Srinivasan, Ruben D Garcia-Ordonez, Patrick R Griffin, Kendall W Nettles |
| National Institutes of Health | T32ES007326 | Julie A Pollock |

| Funder | Grant reference number | Author |
|---|---|---|
| National Institutes of Health | F30DK083899 | Alexander A Parent |
| National Institutes of Health | R01DK101871 | Travis S Hughes, Douglas J Kojetin |
| Florida Department of Health | 1KN-09 | Travis S Hughes, Douglas J Kojetin |
| National Institutes of Health | F32DK097890 | Travis S Hughes |
| National Institute of Diabetes and Digestive and Kidney Diseases (NIDDK) | F32DK097890 | Travis S Hughes |

The funders had no role in study design, data collection and interpretation, or the decision to submit the work for publication.

### Author contributions

JCN, Final approval of the version to be published., Conception and design, Acquisition of data, Analysis and interpretation of data, Drafting or revising the article; SS, TSH, RDG-O, Final approval of the version to be published., Acquisition of data, Analysis and interpretation of data, Drafting or revising the article; NEB, AAP, Final approval of the version to be published., Acquisition of data, Drafting or revising the article; JAP, Final approval of the version to be published., Drafting or revising the article, Contributed unpublished essential data or reagents; OG, VC, Final approval of the version to be published., Acquisition of data, Drafting or revising the article, Contributed unpublished essential data or reagents; JN, RH, Final approval of the version to be published., Acquisition of data, Analysis and interpretation of data, Drafting or revising the article, Contributed unpublished essential data or reagents; PRG, DJK, MDC, Final approval of the version to be published., Analysis and interpretation of data, Drafting or revising the article; JAK, Final approval of the version to be published., Analysis and interpretation of data, Drafting or revising the article, Contributed unpublished essential data or reagents; KWN, Final approval of the version to be published., Conception and design, Analysis and interpretation of data, Drafting or revising the article

# Additional files

## Major datasets

The following datasets were generated:

| Author(s) | Year | Dataset title | Dataset ID and/or URL | Database, license, and accessibility information |
|---|---|---|---|---|
| Nwachukwu JC, Srinivasan S, Bruno NE, Parent AA, Hughes TS, Pollock JA, Gjyshi O, Cavett V, Nowak J, Garcia-Ordonez RD, Houtman R, Griffin PR, Kojetin DJ, Katzenellenbogen JA, Conkright MD, Nettles KW | 2014 | Crystal Structure of the Estrogen Receptor alpha Ligand-binding Domain in Complex with Resveratrol | http://www.rcsb.org/pdb/explore/explore.do?structureId=4pp6 | Publicly available at RCSB protein data bank (www.PDB.org). |
| Nwachukwu JC, Srinivasan S, Bruno NE, Parent AA, Hughes TS, Pollock JA, Gjyshi O, Cavett V, Nowak J, Garcia-Ordonez RD, Houtman R, Griffin PR, Kojetin DJ, Katzenellenbogen JA, Conkright MD, Nettles KW | 2014 | Crystal Structure of the Estrogen Receptor alpha Ligand-binding Domain in Complex with an A-CD ring estrogen derivative | http://www.rcsb.org/pdb/explore/explore.do?structureId=4pps | Publicly available at RCSB protein data bank (www.PDB.org). |
| Nwachukwu JC, Srinivasan S, Bruno NE, Parent AA, Hughes TS, Pollock JA, Gjyshi O, Cavett V, Nowak J, Garcia-Ordonez RD, Houtman R, Griffin PR, Kojetin DJ, Katzenellenbogen JA, Conkright MD, Nettles KW | 2014 | Crystal Structure of the Estrogen Receptor alpha Ligand-binding Domain in Complex with Fluoro-Resveratrol | http://www.rcsb.org/pdb/explore/explore.do?structureId=4ppp | Publicly available at RCSB protein data bank (www.PDB.org). |

The following previously published datasets were used:

| Author(s) | Year | Dataset title | Dataset ID and/or URL | Database, license, and accessibility information |
|---|---|---|---|---|
| Nettles KW, Bruning JB, Gil G, Nowak J, Sharma SK, Hahm JB, Kulp K, Hochberg RB, Zhou H, Katzenellenbogen JA, Katzenellenbogen BS, Kim Y, Joachmiak A, Greene GL | 2008 | Crystal Structure of the Estrogen Receptor Alpha Ligand Binding Domain Mutant 537S Complexed with Genistein | http://www.rcsb.org/pdb/explore/explore.do?structureId=2qa8 | Publicly available at RCSB Protein Data Bank. |
| Brzozowski AM, Pike AC, Dauter Z, Hubbard RE, Bonn T, Engstrom O, Ohman L, Greene GL, Gustafsson JA, Carlquist M | 1997 | Human Estrogen Receptor Ligand-Binding Domain in Complex With 17beta-estradiol | http://www.rcsb.org/pdb/explore/explore.do?structureId=1ere | Publicly available at RCSB Protein Data Bank. |
| Warnmark A, Treuter E, Gustafsson JA, Hubbard RE, Brzozowski AM, Pike ACW | 2002 | Human Oestrogen Receptor Alpha Ligand-binding Domain in Complex with 17beta-oestradiol and TIF2 NRbox3 Peptide | http://www.rcsb.org/pdb/explore/explore.do?structureId=1gwr | Publicly available at RCSB Protein Data Bank. |
| Shiau AK, Barstad D, Loria PM, Cheng L, Kushner PJ, Agard DA, Greene GL | 1998 | Human Estrogen Receptor Alpha Ligand-binding Domain in Complex With 4-Hydroxytamoxifen | http://www.rcsb.org/pdb/explore/explore.do?structureId=3ert | Publicly available at RCSB Protein Data Bank. |

**Standard used to collect data:** PDB standards

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
