## [Decision Letter]

Thank you for sending your work entitled “Resveratrol modulates the inflammatory response via an estrogen receptor-signal integration network” for consideration at *eLife*. Your article has been favorably evaluated by a Senior editor and 3 reviewers, one of whom, Leemor Joshua-Tor, is a member of our Board of Reviewing Editors.

The Reviewing editor and the other reviewers discussed their comments before we reached this decision, and the Reviewing editor has assembled the following comments to help you prepare a revised submission.

This manuscript addresses some of the effects of resveratrol on the ERalpha pathway. The authors claim that much of the attention that has been given to the effects of resveratrol focused on its effect on SIRT1 activation, and given the connection to both pathways and the general interest in resveratrol, the paper would have broad interest. The main previous observation that the authors are examining here is that resveratrol leads to modulation of the inflammatory response but not cell proliferation. Here, the authors present evidence supporting their claim that Resveratrol acts as “as a pathway-selective estrogen receptor-α (ERa) ligand”, a partial agonist that modulates inflammatory responses. The authors describe a series of studies to characterize the ERalpha mediated pathway, which are interesting. The data include a crystal structure of the ERa ligand-binding domain in complex with Resveratrol, gene expression analyses showing that Resveratrol-sensitive TNFa-regulated genes are largely a subset of estrogen-regulated genes, and molecular analyses of IL-6 gene transcription showing ERa dependence for Resveratrol's effects. The reviewers were varied in their opinion as to how compelling these might be.

Issues that should be addressed: 1) In the two-hybrid assay, the authors used 1000 fold different concentrations of E2 vs res. This difference in concentration enhances the conclusion regarding reduced association with SRC1 and SRC3. However, is it really selecting SRC2 at this 1000 fold-higher concentration? In addition, this difference in binding that the authors refer to throughout the manuscript should be shown in vitro to show that this is a direct effect between two components.

2) The authors describe several crystal structures of the LBD of ERalpha with resveratrol and a peptide from SRC2. First, the conclusion from these studies is not apparent. The SRC2 peptide appears to bind a bit differently with resveratrol compared to E2. The authors then state that this is the basis for the specificity difference for SRC2, though the binding difference apparently does not reside in that peptide, but it would change the binding of the region C-terminal to that area. This argument is a bit weak. What are the differences in sequence of the peptides between SRC1,2 & 3 in this region? Are there any? Even if identical, this might still influence the adjacent region, but this is not really explored here sufficiently.

3) In addition, there is some discussion regarding the “dynamic” nature of the ligand binding. This term is used over and over in the manuscript as something unusual or special for this type of ligand binding. But it is not clear what is meant by that. They see the ligand binding in two different orientations. This is static disorder. They haven't shown anything dynamic here. Even the NMR studies do not show this.

4) Many technical details are missing to support the crystallographic and NMR studies:

a. electron density maps are shown without any description as to what type of map is shown, what level are they contoured at etc. The proper maps to show the peptide and ligands should be difference maps prior to these included in the model. 2Fo-Fc maps are prone to bias and are not very informative, if that is what they are showing. What is shown in red, what is shown in green?

b. What are the final occupancies of the two orientations in the final model? How were these come about? On what basis?

c. How many water molecules were included in each structure? What does the Ramachandran map look like? What are the B factors of the ligands and peptides? Why is data completeness low for the first two structures?

d. What does E2 binding look like with the type of NMR experiment that was used?

e. The shift in helices h3 and h12 is not well illustrated. Is the 2.5Å shift of the valine measured with Calpha's? It doesn't look like that big of a shift in the figure.

f. The authors need to comment on whether or not the SRC2 peptide or h12 participate in any lattice (crystal packing) interactions in this structure or in the E2 structure. If so, these conclusions are much weaker.

g. What is meant by h12 dynamics? Again, a snazzy word that is not supported by experiment. This seems like a straight conformational change upon different ligand binding.

6) According to Figure 3, the authors crystallised R with ERalpha/SRC2 peptide and compared it with A-CD ring estrogen complexed with ERalpha/SRC peptide that serves as an agonist control. They saw substantiate changes in the coactivator-binding pocket upon structure overlay. They concluded that this change in the coactivator binding pocket would result in a change in the coactivator binding profile and the lack of proliferative signal. This structure data is not consistent with their biochemical data in Figure 1, middle panel. According to Figure 1, mammalian-2-hybrid showed that there was no change in SRC2 interaction with ERalpha (E2 and R gave similar ∼20 fold luciferase activity). The change in the structure does not translate to what is seen in the cell. Hence the reviewers remain sceptical with the conclusion drawn by the authors from the structure.

7) For Figure 4, what is the basis for the conclusion that “E2 and resveratrol switched SRC2 function from that of a coactivator to a corepressor”? They don't seem to have any effect on SRC2.

8) In Figure 5, the statement regarding dose-dependence rapid increase in NAD+ levels is a bit strong given that only 2 doses were used with just a couple of percent difference for a 10 fold difference. Why is it rapid? In addition, the mechanism by which Resveratrol increases cellular NAD+ levels (Figure 5) is unclear. The authors should clarify.

9) Again, in the Discussion the term “dynamic binding” is used as a mechanism for modulating the inflammatory response without stimulating proliferation, or pathway-selective signaling. No support for “dynamic binding”.

10) Chromatin-bound SIRT1 and PARP-1 have been shown to recruit NMNAT1 to target gene promoters. If they are not involved, what factors mediate the recruitment of NMNAT1 to ERa in response to Resveratrol?

11) The relationship between ERa recruitment and p65 recruitment is not clear. The authors should elaborate in their data, discussion, and model. If TNFa is driving ERa to chromatin, what is the effect of Resveratrol on ERa binding?

12) According to Gehm et al Proc Natl Acad Sci U S A. 1997 December 9; 94(25): 14138-14143 resveratrol was shown to behave as an ER agonist and it stimulated the proliferation of estrogen dependent T47D breast cancer cells. In this work, the authors showed that resveratrol did not bring about cell proliferation in MCF7 cells. However the inflammatory effect of resveratrol is similar in both MCF7 and T47D cells. The authors should discuss this issue in the paper. Otherwise, readers would be mislead into thinking that resveratrol only has an anti-inflammatory response but no effect on breast cell proliferation. The cell proliferative effect seems to be cell line specific. Is this the case? The concluding statement of the Discussion needs to be changed. The authors state at the end of the discussion that 'the work will impact future medicinal chemistry efforts to improve the potency... mediated anti-inflammatory effects while maintaining a neutral profile on ERalpha driven proliferation in the breast”. This sentence is misleading.

13) A tremendous amount of work was done for the silencing experiment. However, the effect of the knockdown is not clean. According to Figure 4—figure supplement 1, the authors were only able to knockdown ∼50% of some of their target. The authors should include data of target protein level to supplement the mRNA level. It is tough to draw clear conclusions from some of the data.

14) If point 11 and 13 are not addressed then the Figure 7 proposed model needs to be re-evaluated to avoid misleading readers.

---

## [Author Response]

*1) In the two-hybrid assay, the authors used 1000 fold different concentrations of E2 vs res. This difference in concentration enhances the conclusion regarding reduced association with SRC1 and SRC3. However, is it really selecting SRC2 at this 1000 fold-higher concentration? In addition, this difference in binding that the authors refer to throughout the manuscript should be shown in vitro to show that this is a direct effect between two components*.

*2) The authors describe several crystal structures of the LBD of ERalpha with resveratrol and a peptide from SRC2. First, the conclusion from these studies is not apparent. The SRC2 peptide appears to bind a bit differently with resveratrol compared to E2. The authors then state that this is the basis for the specificity difference for SRC2, though the binding difference apparently does not reside in that peptide, but it would change the binding of the region C-terminal to that area. This argument is a bit weak. What are the differences in sequence of the peptides between SRC1,2 & 3 in this region? Are there any? Even if identical, this might still influence the adjacent region, but this is not really explored here sufficiently*.

We thank the reviewers for noting the lack of discussion about potential mechanisms for selectivity for different coregulators. This was done in part because the structural features that regulate coregulator assembly represent a very complicated story, which is not central to this manuscript. The complication here is that coactivator recruitment can be driven completely by domains other than the ligand-binding domain, and certainly involves several contacts outside the AF2 surface. These complicated models were described more fully in a review we wrote some time ago (Nettles & Greene *Annu Rev Physiol.* 2005;67:309- 33). We have added an extensive description of these ideas to the beginning of the section describing the structure.

With respect to the different potencies of E2 versus resveratrol, it is important to note that potency of ligands is completely unrelated to their activity profile, or efficacy, as high-affinity ligands can be agonists or antagonists, and the same can be said of low-affinity agonists or antagonists. This is because the efficacy reflects an allosteric effect of the ligand on a second binding site, for transcriptional coactivators. Thus, for assessing efficacy, we simply need to provide a saturating dose of ligand.

We did additional experiments to test if differences in coactivator preferences in the mammalian two-hybrid assay are also apparent with ChIP assays of native proteins, at a gene required for ERα-mediated proliferation. We examined the ligand-induced recruitment of endogenous SRCs to the regulatory region of the *GREB1* gene (Figure 1), and we found the same pattern of coactivator preferences as we saw in the mammalian two-hybrid assay.

We have also added Figure 4 (and Figure 4—figure supplement 1) to further address whether the changes we saw in the crystal structure occur in solution, and if the differences in selectivity occur with the peptides, or require full-length proteins. As requested, we have added the sequences of the peptides to Figure 4—figure supplement 1. Our new data includes FRET assays for determining EC50s of peptide binding, and finally we also added a peptide array experiment that characterized binding of over 150 different peptides. Collectively, these data demonstrate that the AF2 surface has an altered shape in solution, but that the determinants of selectivity for the different SRCs involve full-length proteins.

*3) In addition, there is some discussion regarding the “dynamic” nature of the ligand binding. This term is used over and over in the manuscript as something unusual or special for this type of ligand binding. But it is not clear what is meant by that. They see the ligand binding in two different orientations. This is static disorder. They haven't shown anything dynamic here. Even the NMR studies do not show this*.

From Fundamental’s of Crystallography, Carmelo Giacovazzo, editor, p474: “Another type of disorder may occur in crystals: it is referred to as static disorder, because it is not caused by atomic or molecular motion, but is the result of a number of different atomic dispositions (all corresponding to minima of the packing energy) statistically distributed among different unit cells in the crystal” and “Because of thermal motion, atoms oscillate around an equilibrium position and we may say that they are affected by a small dynamic disorder. But, more often than one might think, larger movements occur in crystals, giving rise to a real dynamic disorder.”

Based on these definitions, the fact that the multiple binding poses occur in solution precludes this from being static disorder. We use “dynamic ligand binding” as a term to describe the fact that the different binding modes are also inducing unique conformers of the receptors. This implies that a dynamically binding ligand provides a mixed population of receptor conformers, where each receptor can undergo a conformational change as the receptor rebinds the ligand, alternates between these conformers. This is the dynamics to which we are referring. We have published two papers in Nature Chemical Biology describing this phenomenon and using this phrase (Srinivasan et al. *Nat Chem Biol*. 2013 May;9(5):326-32; Bruning et al *Nat Chem Bio* 2010; 6(11), 837-843).

Also, please note that this phenomenon has since been described by others (Carroll et al. *J Am Chem Soc.* 2011 Apr 27;133(16):6422-8; Hughes et al. *Structure*. 2012 Jan 11;20(1):139-50), and by a recent finding that extends our observations to GPCRs titled “Dynamic ligand binding dictates partial agonism at a G protein coupled receptor” (Bock et al. *Nat Chem Biol*. 2014 Jan;10(1):18-20). We have amended the text to make the meaning clearer.

*4) Many technical details are missing to support the*
*crystallographic and NMR studies:*

*a. electron density maps are shown without any description as to what type of map is shown, what level are they contoured at etc. The proper maps to show the peptide and ligands should be difference maps prior to these included in the model. 2Fo-Fc maps are prone to bias and are not very informative, if that is what they are showing. What*
*is shown in red, what is shown in green?*

We have revised Figure 3—figure supplement 1 to address this concern. In Figure 3—figure supplement 1, the 2Fo-Fc maps (blue) were contoured at 1σ, while Fo-Fc difference maps (red and green) were contoured at 3σ to indicate where the model is wrong. Red indicates clashes, while green indicates omissions.

The electron densities that were observed in the ligand-binding pockets after molecular replacement and automated rebuilding with Autobuild, and before ligand docking (Figure 3—figure supplement 1); after docking and refinement with the obvious ligand conformer (Figure 3—figure supplement 1), or both ligand conformers (Figure 3—figure supplement 1); and upon shaking coordinates by 1Å, and refinement with simulated annealing after removal of a ligand conformer (Figure 3—figure supplement 1). This was a bias removal procedure developed in collaboration with the PHENIX software structure determination group, and published previously (Bruning et al *Nat Chem Bio* 2010; 6(11), 837-843). The starting model used for molecular replacement already contained the SRC2 peptides in all three cases. Figure 3—figure supplement 1 show the electron density maps for the peptides obtained from the Resveratrol- and A-CD ring estrogen-bound structures, upon shaking coordinates by 1Å, and refinement with simulated annealing, with or without removing the peptide.

*b. What are the final occupancies of the two orientations in the final model? How were these come about? On*
*what basis?*

In the resveratrol structure, both binding orientations fit in the A chain. The occupancy for conformer #1 is about 57%, and that of conformer #2 is about 43%. The B chain mainly fits conformer #2. In the F-resveratrol structure, the A chain fits conformer #1 with about 61% occupancy and conformer #2 with about 39% occupancy, while the B chain mainly fits conformer #2. These values were obtained upon serial occupancy refinement using PHENIX, and have been indicated in the manuscript (Figure 3 & Figure 3—figure supplement 3).

*c. How many water molecules were included in each structure? What does the Ramachandran map look like? What are the B factors of the ligands and peptides? Why is data completeness*
*low for the first two structures?*

The requested information is provided in the revised data collection and refinement statistics table (Table 1). The data completeness is lower at the highest resolution shells of the resveratrol and F-resveratrol-bound structures, consistent with the fewer unique reflections observed in these shells, and the recent changes to recommendations on data processing to include higher resolution data. Moreover, the overall completeness was adequate to solve these structures by molecular replacement.

*d. What does E2 binding look like with the type of NMR*
*experiment that was used?*

We have previously shown that constrained ligands typically show *sharp* peaks while closely related dynamic ligands show *asymmetrical, broader* peaks, or two peaks in these experiments (Srinivasan et al. Nat Chem Biol. 2013 May;9(5):326-32). E2 does not have a Fluorine atom that would enable F19-NMR studies. To address the issue of a control compound, we compared two sets of isomers in our previous publication, of which one set could bind in multiple orientations, whereas their isomers could not. These referenced experiments show matching crystallographic and F19-NMR data on the controls that bind in only one orientation, showing a single sharp peak, while the dynamic ligands showed two peaks or a characteristically broadened peak best modeled as two overlapping peaks.

*e. The shift in helices h3 and h12 is not well illustrated. Is the 2.5Å shift of the valine measured with Calpha's? It doesn't look like that big of a shift in the figure*.

The shift in the alpha carbon is 1Å, the shift in the beta carbon is 1.5Å, and the shift in the gamma carbons is 2.5Å. Thus, this represents a rotation of the side-chain around the main chain to avoid clashing with the phenol, rather than a direct effect on the main chain positioning.

*f. The authors need to comment on whether or not the SRC2 peptide or h12 participate in any lattice (crystal packing) interactions in this structure or in the E2 structure. If so, these conclusions are much weaker*.

Structures superposed in this study have the same space group and show the same crystal packing. H12 does not participate in crystal packing. In contrast, the SRC2 peptide participates in the same crystal packing in both cases. Therefore, the difference in SRC2 positioning is despite being held in place by crystal packing.

*g. What is meant by h12 dynamics? Again, a snazzy word that is not supported by experiment. This seems like a straight conformational change upon different ligand binding*.

“Helix 12 dynamics” refers to the theoretical concept representing the way helix 12 transits through its collection of possible conformational states in solution. This term has been widely used in the nuclear receptor field. The idea here is that a full agonist keeps helix 12 in one conformer, while partial agonists lead it to sample multiple conformers over time, so that the partial activity corresponds to the percent of the time helix 12 is in the active conformer. We have amended the text to explain this better.

*6) According to*
Figure 3*, the authors crystallised R with ERalpha/SRC2 peptide and compared it with A-CD ring estrogen complexed with ERalpha/SRC peptide that serves as an agonist control. They saw substantiate changes in the coactivator-binding pocket upon structure overlay. They concluded that this change in the coactivator binding pocket would result in a change in the coactivator binding profile and the lack of proliferative signal. This structure data is not consistent with their biochemical data in*
Figure 1*, middle panel. According to*
Figure 1*, mammalian-2-hybrid showed that there was no change in SRC2 interaction with ERalpha (E2 and R gave similar ∼20 fold luciferase activity). The change in the structure does not translate to what is seen in the cell. Hence the reviewers remain sceptical with the conclusion drawn by the authors from the structure*.

The difference in SRC2 peptide binding observed in crystal structures is due to a resveratrol-induced change in the shape AF2 surface. As explained in #2 above, this resveratrol-induced structural change generally alters coregulator peptide-binding to the LBD, but is not sufficient to drive the selective reduction of SRC3 recruitment which is observed in the context of full-length or endogenous proteins (Figures 1 and 4). We have added extensive data and discussion to the text to address this issue, highlighting that the selectivity is likely due to communication from the AF2 surface and helix 12 to the other coactivator binding site, AF1, which is located at the amino terminal domain of the receptor.

*7) For*
Figure 4*, what is the basis for the conclusion that “E2 and resveratrol switched SRC2 function from that of a coactivator to a corepressor”? They don't seem to have any effect on SRC2*.

Yes, the ligands do not appear to affect TNFα-induced recruitment of SRC2; however SRC2 *function* appears to be different in the presence of these ligands, based on the effects of SRC2 siRNA compared to control siRNA (Figure 5). SRC2 siRNA causes a ∼2-fold decrease in TNFα-induced expression of *IL-6*, demonstrating that SRC2 is acting to enhance gene expression. In cells transfected with SRC2 siRNA, the ligands were unable to suppress *IL-6* expression, demonstrating that SRC2 is required for repression of *IL-6* by ERα. This has been made clearer in the text. We also now provide references to another paper showing SRC2 has this dual function (Rogatsky et al. *Proc Natl Acad Sci U S A*. 2002 Dec 24;99(26):16701-6), as does the SMRT corepressor (Peterson et al. *Mol Cell Biol*. 2007 Sep;27(17):5933-48).

*8) In*
Figure 5*, the statement regarding dose-dependence rapid increase in NAD+ levels is a bit strong given that only 2 doses were used with just a couple of percent difference for a 10 fold difference. Why is it rapid? In addition, the mechanism by which Resveratrol increases cellular NAD+ levels (*Figure 5*) is unclear. The authors should clarify*.

We agree that testing 2 different doses is not a very strong case for dose-dependency of the increase in NAD+ levels; therefore, we have modified this sentence. Perhaps “transient” is a better description than “rapid”, since we observed this increase within 5 minutes, but not after 10 or 15 minutes (not shown). In other systems, resveratrol is thought to impact NAD+ metabolism through AMPK (Park SJ et al. Cell. 2012 Feb 3;148(3):421-33.), but we do not know how resveratrol increases NAD+ levels in MCF-7 cells.

*9) Again, in the Discussion the term “dynamic binding” is used as a mechanism for modulating the inflammatory response without stimulating proliferation, or pathway-selective signaling. No support for “dynamic binding”*.

Please see #3 above for discussion of the term “dynamic binding” (Carroll et al. *J Am Chem Soc.* 2011 Apr 27;133(16):6422-8; Hughes et al. *Structure*. 2012 Jan 11;20(1):139-50; Bock et al. *Nat Chem Biol*. 2014 Jan;10(1):18-20; Srinivasan et al. *Nat Chem Biol*. 2013 May;9(5):326-32).

*10) Chromatin-bound SIRT1 and PARP-1 have been shown to recruit NMNAT1 to target gene promoters. If they are not involved, what factors mediate*
*the recruitment of NMNAT1 to ERa in response to Resveratrol?*

We concluded that ER mediates recruitment of NMNAT1 to *IL-6* promoter because the ER antagonist, ICI182,780 inhibits recruitment of NMNAT1 to this site (Figure 7). The functional assays suggest that SIRT1 and PARP1 are unlikely to be required for recruitment of NMNAT1 in this context (Figure 5 & Figure 5—figure supplement 1). NMNAT1 does not have an LxxLL motif, and we currently have no clear evidence indicating that ERα binds directly to NMNAT1. Therefore, how ERα mediates recruitment of NMNAT1 to this site is still unknown.

*11) The relationship between ERa recruitment and p65 recruitment is not clear. The authors should elaborate in their data, discussion, and model. If TNFa is driving ERa to chromatin, what is the effect of Resveratrol*
*on ERa binding?*

In MCF-7 cells, resveratrol alone did not induce recruitment of ERα to the *IL-6* promoter (Figure 7—figure supplement 1). In contrast, TNFα alone drove ERα to the *IL-6* promoter (Figure 7). We observed similar effects at the *MCP-1, IL-8, and IL-6* genes in a different MCF-7 sub-line (Nettles et al. *Mol Endocrinol.* 2008 Feb;22(2):263-72).

*12) According to Gehm et al Proc Natl Acad Sci U S A. 1997 December 9; 94(25): 14138-14143 resveratrol was shown to behave as an ER agonist and it stimulated the proliferation of estrogen dependent T47D breast cancer cells. In this work, the authors showed that resveratrol did not bring about cell proliferation in MCF7 cells. However the inflammatory effect of resveratrol is similar in both MCF7 and T47D cells. The authors should discuss this issue in the paper. Otherwise, readers would be mislead into thinking that resveratrol only has an anti-inflammatory response but no effect on breast cell proliferation. The cell proliferative effect seems to be cell line specific. Is this the case? The concluding statement of the Discussion needs to be changed*.

This issue has been extensively addressed in paragraph two of the discussion. There is an extensive body of subsequent literature in both animals and humans that supports a lack of proliferative effect of resveratrol.

*The authors state at the end of the discussion that 'the work will impact future medicinal chemistry efforts to improve the potency... mediated anti-inflammatory effects while maintaining a neutral profile on ERalpha driven proliferation in the breast“. This sentence is misleading*.

We have rewritten the sentence to: “This work advances our understanding of resveratrol action as a modulator of the inflammatory response through ERα, but lacking the feminizing effects of estradiol, and will thus impact future medicinal chemistry efforts to improve the potency or efficacy of resveratrol.”

*13) A tremendous amount of work was done for the silencing experiment. However, the effect of the knockdown is not clean. According to*
Figure 4—figure supplement 1*, the authors were only able to knockdown ∼50% of some of their target. The authors should include data of target protein level to supplement the mRNA level. It is tough to draw clear conclusions from some of the data*.

Unfortunately, this is a caveat of such assays in screening mode, and optimizing western blots for 25 proteins would be a large task. We would like to note that coactivator activity profiles are dosage sensitive. For example, loss of one allele of CBP causes Rubenstein- Taybi syndrome.

*14) If point 11 and 13 are not addressed then the*
Figure 7
*proposed model needs to be re-evaluated to avoid misleading readers*.

The proposed model has been modified as recommended by the reviewers (Figure 8).